# Ecological Land Adaptive Planning in Macroscale, Mesoscale, and Microscale of Shanghai

**Wuyi Jiang [1], Jiawei Xu [2], Yongli Cai [1,3,\*] and Zhiyong Liu [2,\*]** 

[1]   School of Ecological and Environmental Sciences, East China Normal University, Shanghai 200241, China; 52183903007@stu.ecnu.edu.cn
[2]   State Key Laboratory of Radiation Medicine and Protection, School for Radiological and Interdisciplinary Sciences (RAD-X) and Collaborative Innovation Centre of Radiation Medicine of Jiangsu Higher Education Institutions, Soochow University, Suzhou 215123, China; xujiawei@suda.edu.cn
[3]   School of Design, Shanghai Jiaotong University, Shanghai 200240, China
\*   Correspondence: ylcai2020@sjtu.edu.cn (Y.C.); liuzy@suda.edu.cn (Z.L.); Tel.: +86-512-65883945

**Abstract:** The urban ecosystems in China have been compromised during the process of urbanization. The declining services of ecological lands have hindered the sustainable development of cities and the current ecological land management (regulations, rules, and laws) in China cannot meet the demand of future development. In this paper, a new multiscale systematic adaptive ecological land planning method is proposed. Shanghai, a typical mega-city in China, was chosen as the research area. To scientifically and adaptively manage ecological land, downscale management was used and macroscales (city), mesoscales (town), and microscales (community) were chosen. In different scales, different indicators were chosen as evaluation criteria to evaluate the services of the lands. At the mesoscale, habitat quality, carbon sequestration, water conservation, and soil fertility maintenance were chosen. At the mesoscale, habitat quality, carbon sequestration capacity, water production service and food supply were chosen as the evaluation criteria. These indicators are used to evaluate the importance levels of corresponding areas. Based on the importance levels of macroscales and mesoscales, three different scenarios with different targets of Changtian Community were proposed. All three scenarios were judged by stakeholders (residents and managers) of the community and a final scenario was proposed to meet all the requirements. This research not only provides theoretical reference and technical support for ecological land management in different scales of Shanghai, but also provides a new method of adaptive ecological land planning in megacities.

**Keywords:** ecological land; evaluation; classification; adaptive planning; Shanghai

## 1. Introduction

Urbanization is an important issue in the economy development of China. The urban population keeps increasing during the urbanization. In 1979, the urban population made up 19% of the total population (~9.75 billion) and, by 2017, the urban population made up 58.5% of the total population (~13.90 billion) after rapid population growth. Urbanization leads to the great social and economic progresses, but also brings pressure on the environment and ecology. Previous studies [1,2] show that pollutions and contaminations have brought pressure on nature systems and resources and lead to climate change and deterioration of ecological environment, though no evidence shows that economic growth does harm to the natural habitat. In the study of urbanization in China, researchers found that the urbanization has expanded in recent decades but the urbanization levels are still lower than industrialized countries. [3] The construction lands in cities have been expanding during the rapid developing processes, which leads to a disordered land-use situation [2]. Especially in mega cities

(provincial capital cities), the ecological pressure has been loading on the cities because ecological lands, such as forest lands, wetlands, and farmlands, are taking place during the fast expansion of the cities [4–6].

A city is a complicated ecological system, which has a certain complexity and fragility. [7] Compared to normal cities, mega cities are facing much more ecological pressures. The primary ecological problem is the pollution problem. Pollutions such as air pollution, water pollution, soil pollution, and waste discharges are damaging the quality of the whole ecosystem, which directly leads to the living condition problems of human beings [8,9]. Also, the changes in the environment cause the change in the climate, and commonly regional climates worsen during the urbanization [10,11]. Meanwhile, the changes in the ecosystem compromise the cities' ability against natural disasters [12]. Overall, the problems in the ecosystem of urban areas damage the ecological services, which hinders the sustainable development of cities and impact human life [13,14].

To maintain the healthy operation of urban ecosystems, it is highly required to change the land-use methods and enhance the efficiency and serviceability of different lands [15]. Nowadays, China is now in a critical period of social and economic development. The country has come up with a concept of ecological development with high-quality, low energy consumption and ecological environmental protection. In 2015, the government issued a unified plan for the reformation of the whole ecological system, which cleared the boundary between urban and rural areas and proposed classifications and managements of the ecological lands.

However, the management method of ecological land in China is static, coarse, and lacks details of supports and refinements, which brings difficulties during the applications in mesoscale and microscale management. At the meantime, the current concept of ecological land in China is vague and not unified, which is difficult for ecological land-use management. The present ecological land-use management in China is mainly aimed at macroscale, which is city scale, and the coarse management method cannot apply in land-use management in mesoscale and microscale, which are town and community scale, respectively. The present management methods bring the problems of effectiveness of ecological land management.

Therefore, it is necessary to build an adaptive management of ecological land in cities, which suits both the ecosystem level and administrative levels to improve the ecosystem quality and meet the requirements of multiple land uses.

## 2. Study Area, Methods, and Data

### 2.1. Study area

Shanghai (120°51'–122°12' E, 30°40'–31°53' N) is located in the east of China (as shown in Figure 1) and the junction area where the Yangtze River and the Huangpu River meet. As the largest city in China, Shanghai has a population of ~24,237,800 (by the end of 2018) and a land area of 6340 km$^2$. The city's main land-use types are estuarine lands, eastern coastal plain, and western lagoon–marsh deposits area. The city is under the impact of subtropical monsoon climate, which brings ample precipitation. Meanwhile, Shanghai also has large open water areas (697 km$^2$ in total, which accounts for ~11% of the city's area) and lots of rivers run through the city.

In this paper, Liantang Town in Shanghai was selected to further conduct the ecological land planning. Liantang Town is located in the southwest of Shanghai. The area is a low-lying land in the area of the lagoon–marsh deposits and has concentrated lakes and marshes. The town has an area of 92.90 km$^2$ and 25 administrative villages and four communities. Among these, Changtian Community was selected to conduct microscale ecological land-use planning.

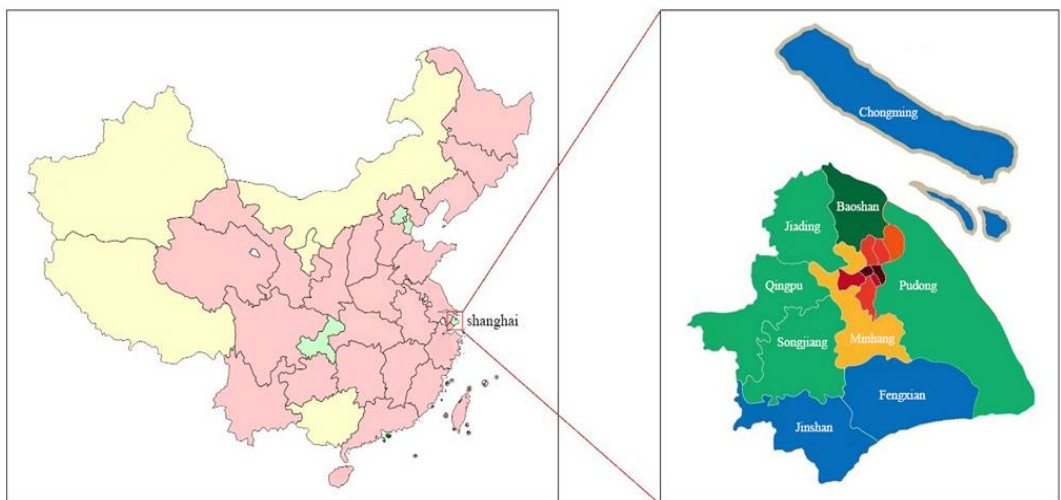

**Figure 1.** Maps of China and Shanghai.

## 2.2. Data and Methods

In this study, land-use evaluation and adaptive planning were applied in three different scales and different evaluating methods and indicators. Most indicators were calculated by the InVEST model. The InVEST (Integrated Valuation of Ecosystem Services and Tradeoffs) model was developed by the Natural Capital Project and has been used in over 60 countries to quantify and value local ecosystem services for decision-makers [16].

### 2.2.1. Macroscale

In macroscale, Level One classification standard of ecological land TM images of Shanghai (of the years 1990, 2000, and 2010) was used to classify the ecological land in the city area. Land-use data of 2018 is from the Resource and Environment Data Center of Chinese Academy of Sciences. Land-use data of these four years all have the resolution of 30 m. The land-use classifications are arable land, forest land, grassland, wetland, and nonecological land. At the meantime, meteorological data, such as precipitation, evaporation, solar radiation data, from 1981 to 2010 are also used. DEM data with the resolution of 30 m were collected from the Geospatial Data Cloud Platform of Computer Network Information Center of Chinese Academy of Sciences. Soil data are from the Harmonized World Soil Database.

To reveal the spatial distribution of ecological land and evaluate the importance level of the land in macroscale, four indicators were chosen, which are carbon sequestration service, water conservation services, habitat quality, and soil erosion. During the carbon sequestration service calculation of the InVEST model, several parameters from previous researches were chosen [17–20] and the carbon densities of different lands are listed in Table 1.

**Table 1.** Carbon densities of different land in Shanghai (t/hm²).

| Land Types | In Aboveground Biomass | In Belowground Biomass | In Soil | In Dead Matter |
|---|---|---|---|---|
| Arable land | 3.87 | 3.27 | 98.30 | 5.00 |
| Forest land | 44.80 | 128.60 | 14.05 | 46.10 |
| Grassland | 35.30 | 86.50 | 99.90 | 2.00 |
| Water area | 0 | 0 | 0 | 0 |
| Construction area | 3.19 | 0.64 | 7.385 | 0 |
| Unconstructed area | 0 | 0 | 18.1 | 0 |

Habitat quality is one of the supporting ecosystem services in InVEST. This indicator quantifies the combination information of land use/land cover (LULC) and threats to biodiversity. It refers to the ecosystem's ability to provide conditions appropriate for individuals and population persistence based on the resources in the ecosystem [15]. Raster data, such as baseline land cover, current land cover, folder containing threat rasters, and sensitivity of land cover types to each threat, were put into the model. Considering the situation in Shanghai, this study took arable land, residential land, industrial and mining land, and transportation land as threats to habitat quality. Meanwhile, forest land, grassland, and water area in the city are taken as habitat. Some parameters of this procedure are shown in Tables 2 and 3, which take previous research [21–23], including studies in China, into consideration [16].

**Table 2.** Threat sources and their maximum influence distance and weight.

| Threats | Maximum Influence Distance | Weight | Type of Decay over Space of the Threats |
|---|---|---|---|
| Arable land | 6 | 0.5 | Linear |
| Residential land | 10 | 1 | Exponential |
| Industrial and mining land | 9 | 0.9 | Exponential |
| Transportation land | 8 | 0.8 | Linear |

**Table 3.** Sensitivity of different habitats in Shanghai to different threat factors.

| Land Types | Habitat Score | Arable Land | Residential Land | Industrial and Mining Land | Transportation Land |
|---|---|---|---|---|---|
| Arable land | 0.6 | 0.2 | 1 | 0.75 | 0.6 |
| Forest land | 0.95 | 0.2 | 0.9 | 0.7 | 0.65 |
| Grassland | 0.8 | 0.6 | 0.3 | 0.4 | 0.52 |
| Water area | 0.9 | 0.1 | 0.9 | 0.8 | 0.4 |
| Construction land | 0.1 | 0.2 | 0.1 | 0.1 | 0.2 |
| Un-used area | 0.3 | 0.5 | 0.6 | 0.3 | 0.3 |

Water conservation in this study used the annual water yield model from the InVEST model to estimate the annual water depth of Shanghai and was modified by topographic index, soil saturated hydraulic conductivity, and flow coefficient. The annual water yield model takes the LULC changes into consideration and the parameters of the model are listed in Table 4, which are combined by previous studies of China [17].

**Table 4.** Parameters of the water conservation model in Shanghai.

| Land-Use Type | Evapotranspiration Index | Root-Restricting Layer Depth | Velocity |
|---|---|---|---|
| Arable land | 0.65 | 300 | 2012 |
| Forest land | 1 | 2000 | 200 |
| Grass land | 0.7 | 700 | 500 |
| Water land | 1 | 1 | 2012 |
| Construction land | 0.01 | 1 | 2012 |
| Unused land | 0.03 | 1 | 500 |

Soil erosion is used to evaluate the soil maintenance in Shanghai and, in this study, RUSLE (Revised Universal Soil Loss Equation) was adopted to calculate soil erosion. The method follows Agriculture Handbook 282 and Agricultural Handbook 537. The parameters in this study also follow previous researches in Shanghai and are listed in Table 5.

| Parameters | Arable Land | Forest Land | Grassland | Water Area | Construction Land | Unused Land |
|:---:|:---:|:---:|:---:|:---:|:---:|:---:|
| C-factor | 0.18 | 0.006 | 0.014 | 0 | 0.20 | 0.06 |
| P-factor | 0.3 | 0.9 | 0.75 | 0 | 0.01 | 0.04 |

The importance level is calculated by spatial analysis function of GIS and the equation is shown as follows:

$$ES_j = 4\sqrt[4]{\prod_{i=1}^{4} S_i} \qquad (1)$$

where $ES_i$ is the comprehensive evaluation index of ecological services of ecological land, and $S_i$ is the normalized value of certain ecological services (carbon sequestration service, water conservation services, habitat quality, and soil erosion). The $ES_i$ values are divided into four different levels, which are extremely important (0.85–1), important (0.75–0.85), generally important (0.65–0.75), and unimportant (0–0.65).

### 2.2.2. Mesoscale

In mesoscale, data with higher resolution are used. Images with 1 m resolution from Gaofen-2 Satellite and basic information of Liantang Town were used. At the meantime, meteorological data, soil data and socioeconomic data are used as auxiliary data. In mesoscale, four indicators are selected to evaluate the basic ecological services of the town, and they are carbon sequestration service, habitat quality, water production service, and food supply service.

The habitat quality service, water production service (water yield), and carbon sequestration service take the same method of macroscale (Section 2.2.1). The food supply service is based on the previous study of Xie et al. [24] to evaluate the grain production capacity of different types of land in Liantang Town, and some of the parameters are modified by local researches, which are listed in Table 6.

**Table 6.** Grain production capacity of unit area in Liantang Town.

| Land Types | Grain Production Capacity | Land Types | Grain Production Capacity |
|:---:|:---:|:---:|:---:|
| Dry land | 1 | Shrub Land | 0.34 |
| Paddy field | 1 | Sparse woodland | 0.35 |
| Agricultural facility land | 0.95 | Grassland | 0.43 |
| Fish pond | 0.65 | River | 0.53 |
| Orchard | 0.6 | Pond | 0.5 |
| Vegetable field | 0.9 | Ditch | 0.49 |
| Forest land | 0.33 | Tidal flat and wetland | 0.36 |

### 2.2.3. Microscale

In microscale, three main types of data are used, which are geographic information data, social information data, and site survey and questionnaire survey data. The geographic information data of Changtian Community includes land-use types, altitudes, et al. Social information data includes population size, number of households, land ownership, and incomes. The site survey is conducted by authors to check the reliability of the data, and the questionnaire survey is mainly to collect the voice of residents and community managers.

### 2.2.4. The Planning System of Multiscale Ecological Land

To show the main process of this study, the flow/system is shown in Figure 2. The detailed processes are presented in the following sections.

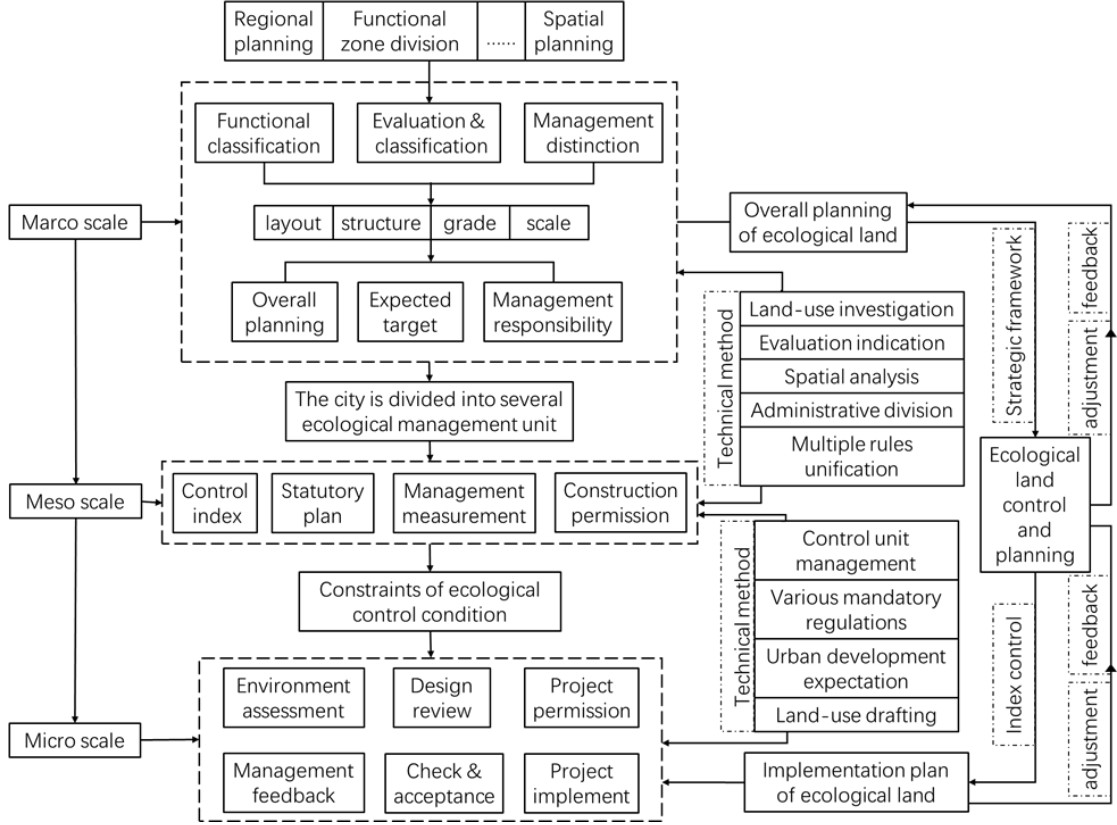

**Figure 2.** Planning system of the multiscale ecological land.

In traditional land management in China, the methods are mostly static, which lacks flexible adaption on local ecological environments, political managements and public demands. Therefore, the adaptive planning and management are proposed in this study to meet the demands of management of different scales and reduce the adverse effects of uncertain factors on urban ecological land. From the whole city scale, the ecological land planning is a complete system that can be executed through different scales and different levels and optimized to meet the requirements of the dynamic changes of cities.

This study is mainly about the ecological land-use planning and management of Shanghai, which focuses on the adaptive planning in multiple scales and multiple levels. The adaptive planning takes the main characteristics of the study area and the differences between different land types into consideration. For example, in the mesoscale study, the grain production supply was chosen as one indicator because Liantang Town is a typical agricultural production area in Shanghai. Overall, spatial scale, administration level, objects et al. should be included in the adaptive planning.

In different spatial scales, the research scope, the main characteristics, and data resolutions vary widely. Therefore, it is necessary to match the ecological land spatial scale with the corresponding administrative scale.

Administrative level is an important factor which influences the direction of the ecological land planning. Administrative level can be divided into decision-making level, management level, and execution level, which are responsible for different duties.

Also, different objects have different management rules. For example, arable land and forest land are under the department of agriculture and forestry, respectively, so that they are under different management rules. Therefore, the adaptive method will be changed due to different objects.

The adaptive method should also consider the future of the ecological land, which includes the appeal of main ecological functions of the ecosystem, the economic development direction, and the decision made by the rule-makers (residents and government).

The detailed results are shown in the following section.

### 3. Processings in Different Scales

*3.1. Evaluation and Planning of Ecological Land Use at City (Macro) Scale*

Figure 3 shows the land use of Shanghai (macroscale) in 2018. Table 7 is the land utilization situation of Shanghai in four different years (1990, 2000, 2010, 2018), and Table 8 is the proportion matrix of land-use transfer in Shanghai from 1990 to 2018. In 2018, the main land-use types of Shanghai were cultivated lands and construction lands, which account for over 90% of the total area. The cultivated area is mainly distributed in the south of the city and Chongming Island. During the period of 1990–2000, arable area decreased by 8.5% and grassland decreased by 60.1%. The constructive area increased significantly, with an increasing rate of 38.7%. From the land transfer matrix, the lost arable land was mostly transferred into constructive area, while 18.9% of the grassland was transferred into constructive area. The matrix also shows that 44% of the grassland was transferred into the water area. During the period of 2000–2010, the constructive areas increased rapidly with a rate of 58.1%. In the meantime, the arable area decreased substantially by 18.6%, and the decreased area was mostly changed into the constructive area. Other types of land (expect unused land) were transferred into constructive areas in different degrees. During the period of 2010–2018, the constructive area still increased with a relatively smaller rate of 21.4%. In this period, grassland increased unprecedentedly with a rate of 506%. The water area and unused land increased as well. Meanwhile, the transfer rates between different types of lands slowed down. In this period, the arable area still was the main source of constructive land and ~14% of the arable area, along with 7.3% of water area, 9.6% of forest land and 25.1% of grassland, changed into constructive land.

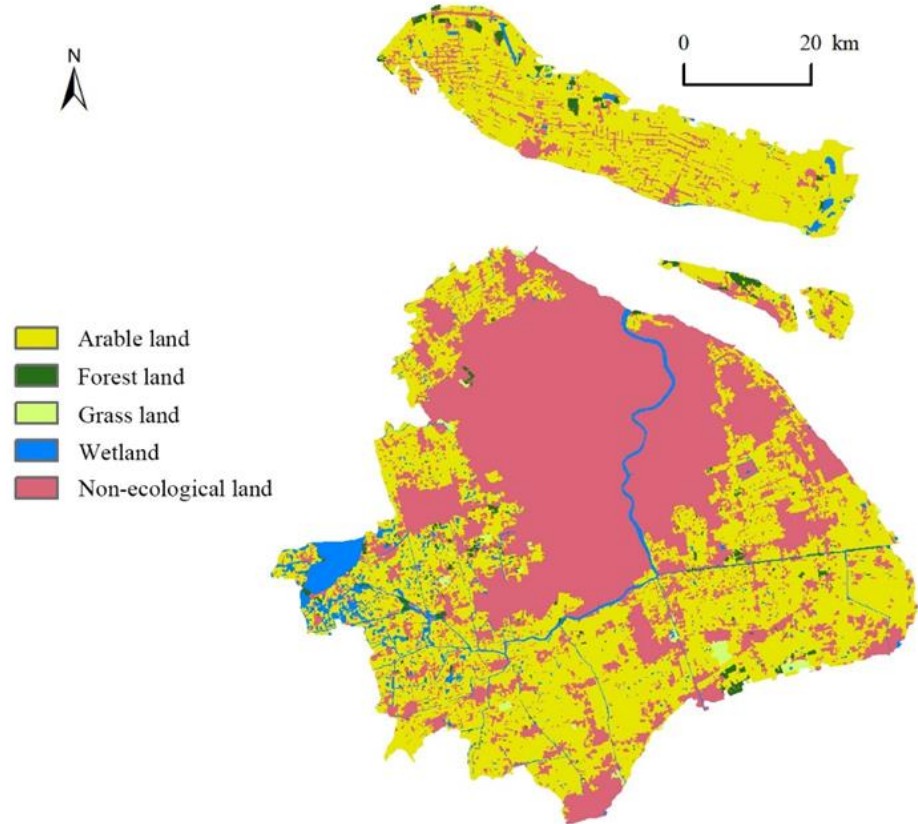

**Figure 3.** Land-use map of Shanghai in 2018.

**Table 7.** Land-use condition of Shanghai of 1990, 2000, 2010, and 2018.

| Land-Use Type | 1990 | | 2000 | | 2010 | | 2018 | |
|---|---|---|---|---|---|---|---|---|
| | Area | Ratio | Area | Ratio | Area | Ratio | Area | Ratio |
| | km² | % | km² | % | km² | % | km² | % |
| Arable land | 4855.91 | 77.98 | 4443.94 | 71.36 | 3616.99 | 58.08 | 3115.53 | 49.98 |
| Forest land | 100.84 | 1.62 | 98.57 | 1.58 | 97.06 | 1.56 | 81.92 | 1.31 |
| Grass land | 16.69 | 0.27 | 6.57 | 0.11 | 5.92 | 0.10 | 35.93 | 0.58 |
| Water area | 217.83 | 3.50 | 241.40 | 3.88 | 235.63 | 3.78 | 238.37 | 3.82 |
| Constructive area | 1035.90 | 16.63 | 1436.67 | 23.07 | 2271.55 | 36.48 | 2760.74 | 44.29 |
| Unused land | 0.34 | 0.01 | 0.34 | 0.01 | 0.34 | 0.01 | 0.59 | 0.01 |

**Table 8.** Matrix of land transfer in Shanghai from 1990–2018 (%).

| Period | Land Type | Arable Land | Forest Land | Grass Land | Water Area | Constructed Area | Unused Area |
|---|---|---|---|---|---|---|---|
| 1990–2000 | Arable land | 91.5 | 0.0 | 0.0 | 0.3 | 8.2 | 0.0 |
| | Forest land | 0.1 | 97.5 | 0.0 | 0.1 | 2.3 | 0.0 |
| | Grass land | 0.1 | 0.0 | 37.0 | 44.0 | 18.9 | 0.0 |
| | Water area | 0.1 | 0.0 | 0.1 | 99.7 | 0.1 | 0.0 |
| | Constructed area | 0.1 | 0.0 | 0.0 | 0.0 | 99.9 | 0.0 |
| | Unused area | 0.0 | 0.0 | 0.0 | 0.3 | 0.3 | 99.5 |
| 2000–2010 | Arable land | 81.3 | 0.1 | 0.0 | 0.3 | 18.3 | 0.0 |
| | Forest land | 0.0 | 90.4 | 0.0 | 0.2 | 9.3 | 0.0 |
| | Grass land | 9.4 | 29.8 | 56.5 | 2.1 | 2.1 | 0.0 |
| | Water area | 2.3 | 0.3 | 0.9 | 91.5 | 4.9 | 0.0 |
| | Constructed area | 0.0 | 0.0 | 0.0 | 0.0 | 100.0 | 0.0 |
| | Unused area | 0.0 | 0.0 | 0.0 | 0.0 | 0.0 | 100.0 |
| 2010–2018 | Arable land | 84.5 | 0.0 | 0.6 | 0.6 | 14.1 | 0.0 |
| | Forest land | 2.4 | 82.7 | 4.0 | 1.3 | 9.6 | 0.0 |
| | Grass land | 11.6 | 0.0 | 49.7 | 13.5 | 25.1 | 0.0 |
| | Water area | 4.3 | 0.2 | 0.3 | 87.9 | 7.3 | 0.0 |
| | Constructed area | 1.8 | 0.0 | 0.2 | 0.2 | 97.8 | 0.0 |
| | Unused area | 0.0 | 0.0 | 0.0 | 100.0 | 0.0 | 0.0 |
| 1990–2018 | Arable land | 63.6 | 0.1 | 0.6 | 0.9 | 34.8 | 0.0 |
| | Forest land | 2.7 | 75.4 | 0.4 | 1.3 | 20.1 | 0.0 |
| | Grass land | 18.5 | 2.0 | 24.6 | 15.6 | 39.3 | 0.0 |
| | Water area | 4.4 | 0.3 | 0.5 | 84.7 | 10.2 | 0.0 |
| | Constructed area | 0.8 | 0.0 | 0.1 | 0.3 | 98.9 | 0.0 |
| | Unused area | 0.5 | 0.0 | 0.0 | 99.5 | 0.0 | 0.0 |

From 1990 to 2018, the areas of arable land and forest land decreased by 35.9% and 18.8%, respectively. Constructive area, grassland, water area, and unused land increased by 166.3%, 115.1%, 9.3%, and 72%, respectively. In this period, 35% of arable area, 20% of forest land, and 40% of grassland have been transferred into constructive areas, which results in the continuous increase of constructive area in about 30 years.

In the spatial distribution of carbon sequestration (Figure 4), southern Shanghai and Chongming Island have better ability of carbon sequestration, while the downtown and its surrounding area have lower carbon sequestration. With the expansion of the urban areas, the low carbon sequestration area spreads from 1990 to 2018. All the districts of Shanghai show an overall decrease in carbon sequestration (Table 9). During the period, Chongming Island, Pudong District, and Fengxian District have higher carbon sequestration, with average carbon sequestrations of 11.39 Tg, 9.17 Tg, and 6.26 Tg. The downtown area has the lowest carbon sequestration of all years, with an average of 0.4 T. For the carbon sequestration per area, Chongming Island, Fengxian District, and Qingpu District have the highest values, which are 9.05 t/km², 6.94 t/km², and 6.8 t/km² respectively. The downtown area still has the lowest carbon sequestration per area, with a value of 1.25 t/km².

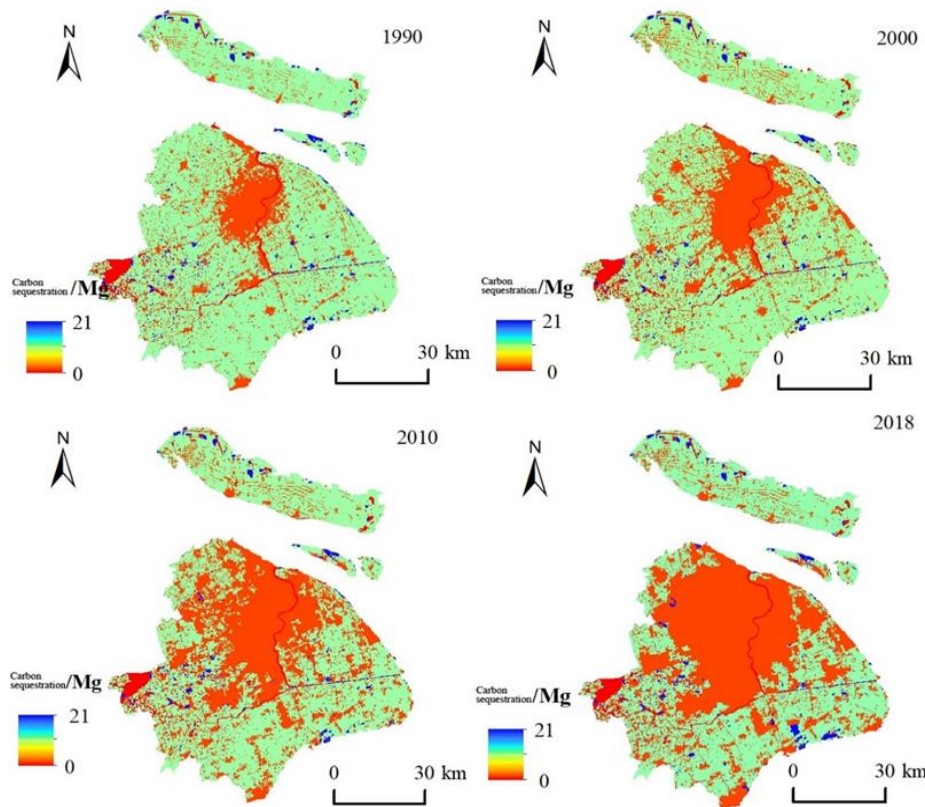

**Figure 4.** Carbon Sequestration Service of ecosystem in Shanghai from 1990 to 2018 (Mg).

**Table 9.** Carbon sequestration of districts in Shanghai from 1990 to 2018 (Tg).

| District | 1990 | 2000 | 2010 | 2018 |
|---|---|---|---|---|
| Baoshan | 2.13 | 1.59 | 0.98 | 0.58 |
| Chongming | 12.00 | 11.40 | 11.13 | 11.01 |
| Fengxian | 6.70 | 6.65 | 5.86 | 5.83 |
| Jiading | 4.36 | 3.87 | 2.71 | 1.97 |
| Jinshan | 5.75 | 5.68 | 5.04 | 4.83 |
| Minhang | 2.90 | 2.43 | 1.49 | 0.89 |
| Pudong | 11.44 | 10.15 | 8.32 | 6.78 |
| Qingpu | 5.72 | 5.47 | 4.68 | 4.15 |
| Downtown | 0.61 | 0.35 | 0.33 | 0.31 |
| SongJiang | 5.90 | 5.54 | 4.35 | 3.87 |

The correlations between the carbon sequestration and different land types in Shanghai in four different years are shown in Table 10. As seen in the table, lands with high carbon density, such as arable land and forest land, have significant high correlations with carbon sequestrations. Arable land and forest land are the areas with high carbon density, which slow down the losing rate of carbon sequestration in Shanghai. Though grassland has high carbon density, the area of grassland is too small to influence the carbon sequestration in the ecosystem. The constructive area has significant negative correlations with carbon sequestration in the period ($-0.822$, $-0.915$, $-0.556$, $-0.911$). The increasing area of constructive land in Shanghai leads to loss of carbon sequestration directly.

In the urbanization progress of Shanghai, the area of ecosystem is shrinking continuously, which compromises the carbon sequestration ability of Shanghai. The loss of carbon sequestration in Shanghai is contributed to expansion of constructive land and compromise of arable land. Also, due to the strict farmland protection regulations, the compensation of arable land is from forest land and grassland, which also leads to the reduction of the carbon sequestration level of the ecosystem.

**Table 10.** Correlations between different land types and carbon sequestration in different years.

| Land Type | 1990 | 2000 | 2010 | 2018 |
|---|---|---|---|---|
| Arable land | 0.660 * | 0.802 ** | 0.511 * | 0.827 ** |
| Forest land | 0.684 * | 0.636 * | 0.629 * | 0.719 ** |
| Grass land | 0.387 | 0.314 | −0.024 | 0.405 |
| Water area | 0.074 | −0.503 | −0.135 | 0.084 |
| Constructed area | −0.822 ** | −0.915 ** | −0.556 * | −0.911 ** |
| Unused area | −0.029 | −0.089 | 0.122 | 0.267 |

\* Correlation is significant at the 0.01 level (2-tailed); \*\*Correlation is significant at the 0.05 level (two-tailed).

The habitat quality results of four years are shown in Figure 5, and the detailed habitat qualities of each district in Shanghai are shown in Table 11. Habitat quality ranges from 0 to 1, and the values closer to 1 show the better habitat quality of the areas. The better habitat quality indicates that the city has a relatively complete ecosystem, which can maintain the biodiversity of the area. The average habitat quality of Shanghai in 2018 is 0.35, which is the lowest value from 1990. Habitat qualities of different districts in Shanghai in four different years are shown in Table 11 and habitat qualities of different land types are shown in Table 12. The downtown area has the worst habitat quality and the area expands through the years. In the year of 1990, Qingpu District, Chongming Island, Fengxian District, and Jinshan District were the four district areas having the highest habitat quality values. With the development of the city, Chongming Island, Qingpu District, and Jinshan District are the three districts with the highest habitat qualities in the year of 2018. The results also indicate that Chongming Island is the least influenced district in Shanghai, while Minhang District is the most influenced district by the urbanization.

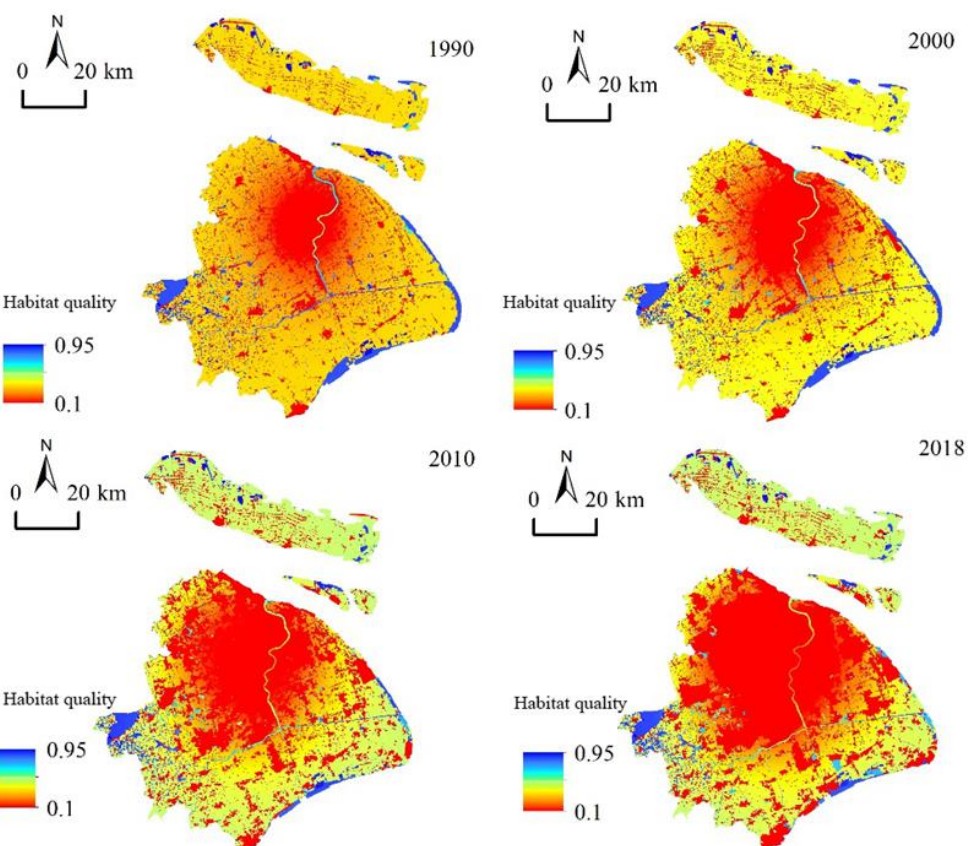

**Figure 5.** Habitat quality distribution of Shanghai from 1990 to 2018.

**Table 11.** Habitat qualities of districts and Shanghai in 1990, 2000, 2010, and 2018.

| District | 1990 | 2000 | 2010 | 2018 | Average |
|---|---|---|---|---|---|
| Baoshang | 0.38 | 0.29 | 0.17 | 0.13 | 0.24 |
| Chongming | 0.58 | 0.56 | 0.54 | 0.53 | 0.55 |
| Fengxian | 0.55 | 0.54 | 0.45 | 0.42 | 0.49 |
| Jiading | 0.50 | 0.44 | 0.29 | 0.22 | 0.36 |
| Jinshang | 0.55 | 0.54 | 0.47 | 0.45 | 0.51 |
| Minhang | 0.41 | 0.34 | 0.20 | 0.15 | 0.27 |
| Pudong | 0.51 | 0.44 | 0.35 | 0.29 | 0.40 |
| Qingpu | 0.60 | 0.57 | 0.49 | 0.44 | 0.52 |
| Downtown | 0.16 | 0.13 | 0.12 | 0.11 | 0.13 |
| Songjiang | 0.54 | 0.50 | 0.37 | 0.33 | 0.44 |
| Shanghai | 0.51 | 0.47 | 0.39 | 0.35 | 0.43 |

**Table 12.** Habitat quality index of different land types in Shanghai.

| | 1990 | | 2000 | | 2010 | | 2018 | |
|---|---|---|---|---|---|---|---|---|
| Land Type | Habitat Quality | Change Ratio (%) | Habitat Quality | Change Ratio (%) | Habitat Quality | Change Ratio (%) | Habitat Quality | Change Ratio (%) |
| Arable land | 0.58 | 87.45 | 0.56 | 84.88 | 0.53 | 79.15 | 0.53 | 74.41 |
| Forest land | 0.93 | 2.93 | 0.92 | 3.06 | 0.87 | 3.47 | 0.86 | 3.18 |
| Grass land | 0.79 | 0.41 | 0.78 | 0.17 | 0.78 | 0.19 | 0.77 | 1.26 |
| Water area | 0.88 | 5.97 | 0.86 | 7.03 | 0.82 | 7.91 | 0.80 | 8.67 |
| Constructed area | 0.10 | 3.23 | 0.10 | 4.85 | 0.10 | 9.28 | 0.10 | 12.47 |
| Unused area | 0.89 | 0.01 | 0.88 | 0.01 | 0.87 | 0.01 | 0.72 | 0.02 |

In nearly 30 years, the massive-scale human activities in Shanghai had led to an increase in land-use intensity and weakened the quality of the ecological environment in Shanghai. Especially, in the downtown area, the population density and construction density are much higher than other districts of Shanghai, which leads to the declining quality of the ecosystem.

The water conservation index (indicator) is mainly measured by precipitation interception, runoff regulation, evapotranspiration, water purification, etc. The results of water conservation are listed in Table 13 and shown in Figure 6. From Table 13, water conservation in all districts of Shanghai shows an overall decreasing trend from 1990 to 2018 and in 2018, Shanghai has a water yield of $7.54 \times 10^8$ t. The average water depth decreases 22.52 mm and Baoshan District has the largest decrease in water depth, which is 62.79 mm. Meanwhile, Chongming Island lost 8.9 mm in water depth during ~30 years, which indicates its important role in maintaining the water conservation service in Shanghai. By the end of 2018, the districts with water depths over 120 mm are only Jinshan District, Fengxian District, Qingpu District, Songjiang District, and Chongming Island. From the spatial distribution (Figure 6), the water conservation index decreases gradually from north to south. Jinshan District, Qingpu District, Songjiang District, Fengxian District, and Chongming Island have larger water conservation indexes.

**Table 13.** Water conservation (water depth and yield) of districts in Shanghai from 1990 to 2018.

| District | 1990 | | 2000 | | 2010 | | 2018 | |
|---|---|---|---|---|---|---|---|---|
| | Depth (mm) | Yield ($10^8$t) | Depth (mm) | Yield ($10^8$t) | Depth (mm) | Yield ($10^8$t) | Depth (mm) | Yield ($10^8$t) |
| Baoshan | 107.53 | 0.34 | 86.05 | 0.27 | 61.27 | 0.19 | 44.74 | 0.14 |
| Chongming | 133.13 | 1.53 | 128.69 | 1.48 | 125.62 | 1.44 | 124.34 | 1.43 |
| Fengxian | 171.69 | 1.24 | 170.66 | 1.23 | 156.32 | 1.13 | 154.01 | 1.11 |
| Jiading | 128.53 | 0.65 | 116.83 | 0.59 | 87.83 | 0.44 | 68.25 | 0.34 |
| Jinshan | 201.46 | 1.26 | 200.01 | 1.25 | 185.93 | 1.16 | 181.28 | 1.14 |
| Minhang | 124.46 | 0.52 | 110.84 | 0.46 | 81.86 | 0.34 | 63.01 | 0.26 |
| Pudong | 139.72 | 1.82 | 127.79 | 1.67 | 110.19 | 1.44 | 95.59 | 1.25 |
| Qingpu | 147.82 | 1.08 | 143.54 | 1.05 | 129.89 | 0.95 | 120.44 | 0.88 |
| Downtown | 54.82 | 0.18 | 44.71 | 0.14 | 43.73 | 0.14 | 42.90 | 0.14 |
| Songjiang | 165.26 | 1.11 | 158.61 | 1.07 | 135.73 | 0.91 | 126.16 | 0.85 |
| Shanghai | 143.86 | 9.72 | 136.23 | 9.21 | 120.58 | 8.15 | 111.46 | 7.54 |

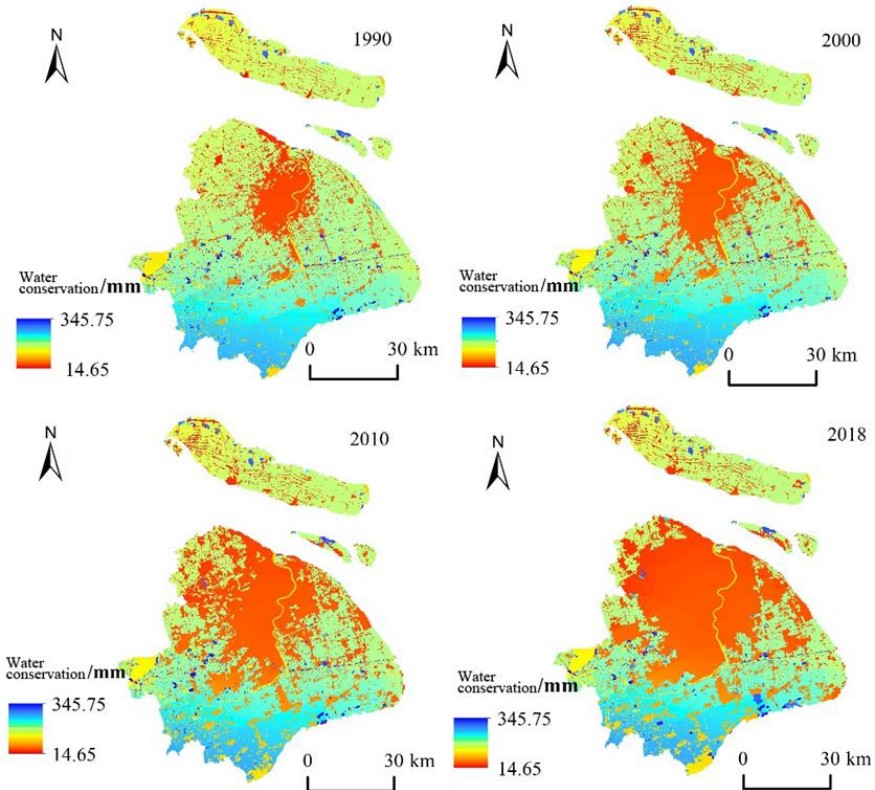

**Figure 6.** Water conservation service in Shanghai from 1990 to 2018 (mm).

The water conservation services of different land types change due to human activities. Therefore, water conservation services of different lands in Shanghai in the last three decades are calculated and listed in Table 14. As seen in the table, forest land is the one with the highest water depth per area (276.76 mm). Grassland comes second, and arable land comes third, with 226.82 mm and 166.84 mm water depth, respectively. The loss of ecological land (forest land, grassland, etc.) is the main reason for the decreasing water conservation service. The correlations between land types and water conservation service indexes are listed in Table 15. The table shows that forest land and grassland have significant positive correlations with water conservation during the period and the primary reason is that the plant layer can effectively intercept precipitation, increasing moisture in the soil and promoting the ability of water conservation. The reason for the low correlation between constructed land and water conservation service is that construction land destroys the structure of soil and the compaction and crusting of soil make the water content in soil decrease continuously. Overall, the changes in land-use influence the water capacity of the underlying surface and then influence the water conservation service in the city. If the worsening of water conservation services continues, the water cycle in nature would change and endanger the security of ecosystem in Shanghai.

**Table 14.** Water conservation of ecosystem in different land types from 1990 to 2018.

| Land Type | 1990 | | 2000 | | 2010 | | 2018 | |
|---|---|---|---|---|---|---|---|---|
| | Depth (mm) | Yield ($10^8$t) | Depth (mm) | Yield ($10^8$t) | Depth (mm) | Yield ($10^8$t) | Depth (mm) | Yield ($10^8$t) |
| Arable land | 163.93 | 86.45 | 164.88 | 79.50 | 165.20 | 64.55 | 165.88 | 55.63 |
| Forest land | 275.02 | 2.96 | 275.15 | 2.89 | 275.10 | 2.85 | 273.56 | 2.37 |
| Grass land | 211.09 | 0.21 | 215.09 | 0.11 | 213.31 | 0.10 | 238.49 | 0.88 |
| Water area | 109.32 | 2.53 | 108.20 | 2.69 | 109.40 | 2.71 | 110.77 | 2.79 |
| Constructed area | 44.91 | 5.11 | 43.69 | 6.90 | 45.28 | 11.31 | 45.20 | 13.72 |
| Unused area | 157.40 | 0.00 | 157.89 | 0.00 | 157.89 | 0.00 | 157.68 | 0.01 |

**Table 15.** Correlations between land types and water conservation in Shanghai from 1990 to 2018.

| Land Type | 1990 | 2000 | 2010 | 2018 |
|:---:|:---:|:---:|:---:|:---:|
| Arable land | 0.467 | 0.109 | 0.272 | 0.432 |
| Forest land | 0.808 ** | 0.733 ** | 0.703 ** | 0.773 ** |
| Grass land | 0.511 * | 0.544 ** | 0.652 * | 0.687 * |
| Water area | 0.244 | 0.363 | 0.399 | 0.207 |
| Constructed area | −0.600 * | −0.719 * | −0.696 * | −0.815 * |
| Unused area | 0.218 | 0.193 | 0.382 | 0.445 |

* Correlation is significant at the 0.01 level (2-tailed); **Correlation is significant at the 0.05 level (2-tailed).

The soil erosion in Shanghai was analyzed and Figure 7 is the spatial distribution of soil erosion modules. As shown in the figure, the downtown area and its surrounding area have a less amount of soil erosion, while the south of the city has a large amount of soil erosion. The area with less soil erosion expands through the period. Table 16 lists the soil erosion amounts of all the districts in Shanghai. The total amount of soil erosion of Shanghai reduced from $1120.63 \times 10^4$ t to $743.17 \times 10^4$ t, which indicates that the soil maintenance improved during this period.

Table 17 shows the soil erosion amount of different types of lands. From 1990 to 2018, arable land provided ~95% of soil erosion, while forest provided the least soil erosion. The lost area of the arable area might contribute to the decreasing trend of soil erosion in the study period. Also, the districts with larger areas of arable lands, such as Chongming Island, Pudong District, and Fengxian District, have larger annual soil erosion amount. The arable land provides food to the local community, and converting farmlands into forests in the premise that food is sufficient for the community is suggested.

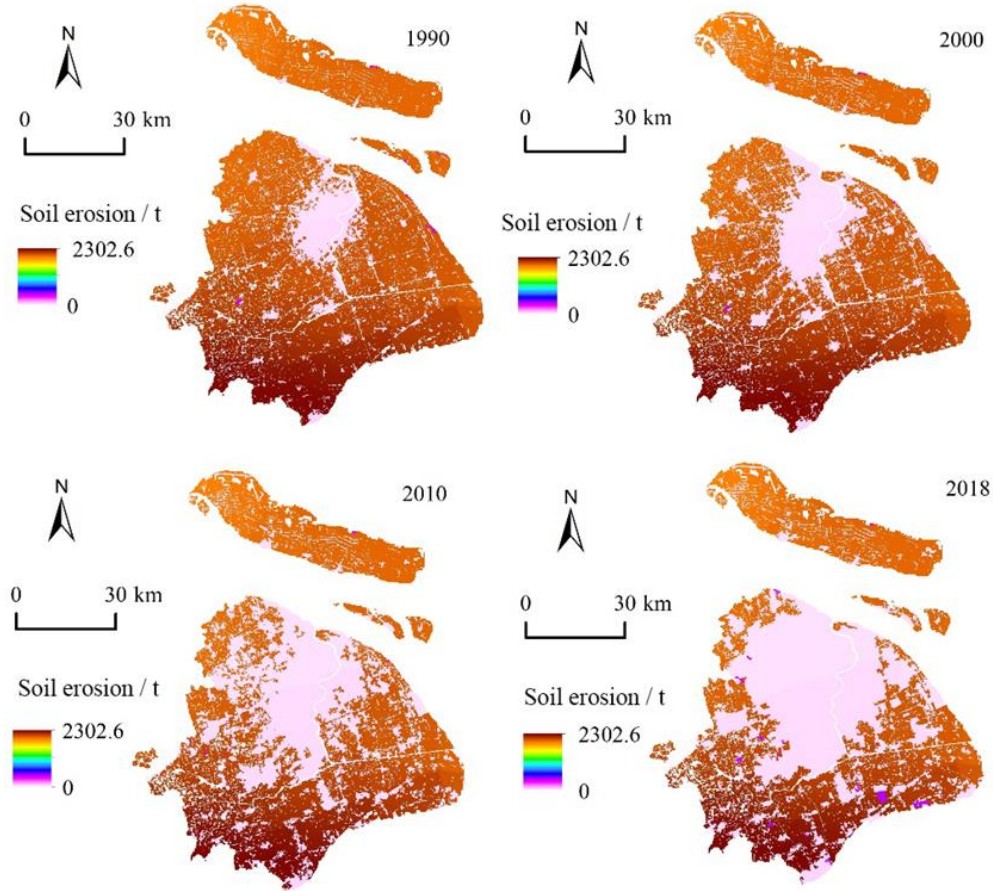

**Figure 7.** Soil erosion modules in Shanghai from 1990 to 2018 (t/km$^2$·a).

**Table 16.** Soil erosion modules ($10^4$ t) and ratios of districts in Shanghai from 1990 to 2018.

| District | 1990 Total Soil Erosion | % | 2000 Total Soil Erosion | % | 2010 Total Soil Erosion | % | 2018 Total Soil Erosion | % |
|---|---|---|---|---|---|---|---|---|
| Baoshan | 42.47 | 3.79 | 30.76 | 2.98 | 17.58 | 2.07 | 8.20 | 1.10 |
| Chongming | 205.38 | 18.33 | 196.42 | 19.03 | 190.75 | 22.41 | 188.27 | 25.33 |
| Fengxian | 137.74 | 12.29 | 136.46 | 13.22 | 118.72 | 13.95 | 110.84 | 14.91 |
| Jiading | 87.28 | 7.79 | 77.15 | 7.47 | 51.38 | 6.04 | 34.14 | 4.59 |
| Jinshan | 127.10 | 11.34 | 125.54 | 12.16 | 110.52 | 12.99 | 103.73 | 13.96 |
| Minhang | 57.23 | 5.11 | 47.10 | 4.56 | 27.02 | 3.17 | 14.77 | 1.99 |
| Pudong | 221.06 | 19.73 | 195.31 | 18.92 | 157.33 | 18.48 | 126.38 | 17.01 |
| Qingpu | 112.68 | 10.06 | 107.00 | 10.37 | 89.40 | 10.50 | 78.68 | 10.59 |
| Downtown | 10.06 | 0.90 | 4.47 | 0.43 | 3.93 | 0.46 | 3.55 | 0.48 |
| Songjiang | 119.63 | 10.67 | 112.00 | 10.85 | 84.46 | 9.92 | 74.60 | 10.04 |
| Shanghai | 1120.63 | | 1032.22 | | 851.10 | | 743.17 | |

**Table 17.** Total soil erosion ($10^4$) and ratios (%) of different land types in Shanghai from 1990 to 2018.

| Land Type | 1990 Total Soil Erosion | % | 2000 Total Soil Erosion | % | 2010 Total Soil Erosion | % | 2018 Total Soil Erosion | % |
|---|---|---|---|---|---|---|---|---|
| Arable land | 1106.37 | 98.72 | 1013.12 | 98.14 | 821.34 | 96.49 | 705.73 | 94.95 |
| Forest land | 0.75 | 0.07 | 0.73 | 0.07 | 0.72 | 0.08 | 0.60 | 0.08 |
| Grass land | 0.46 | 0.04 | 0.24 | 0.02 | 0.22 | 0.03 | 1.77 | 0.24 |
| Water area | 0.00 | 0.00 | 0.00 | 0.00 | 0.00 | 0.00 | 0.00 | 0.00 |
| Constructed area | 13.17 | 1.18 | 18.23 | 1.77 | 28.91 | 3.40 | 35.13 | 4.73 |
| Unused area | 0.01 | 0.00 | 0.01 | 0.00 | 0.01 | 0.00 | 0.04 | 0.01 |

Based on the four indicators above, the indicators of 2018 are chosen and normalizations of these four indicators are made to determine the importance level of the ecosystem services in Shanghai (Figure 8). The normalization results are divided into four ranks in Shanghai (extremely important, important, generally important, and unimportant) and the results are shown in Figure 9. The important and unimportant ecological land covers over 80% of the area in Shanghai, while extremely important ecological land covers less than 4% of the total area.

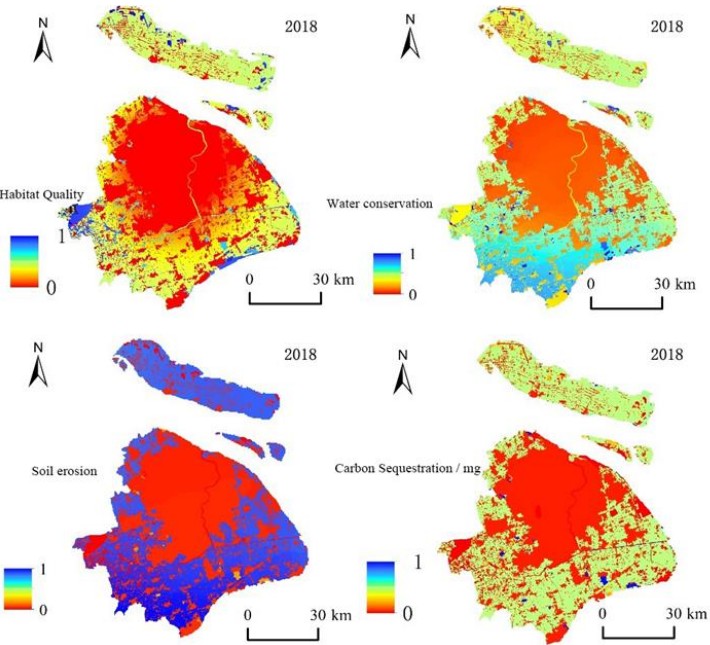

**Figure 8.** Normalization of ecosystem services in Shanghai.

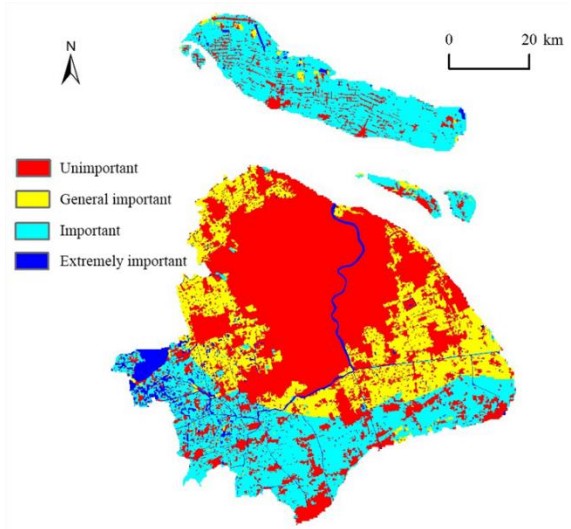

**Figure 9.** Importance level of ecosystem services in Shanghai.

According to the importance level of ecosystem services, it can be seen that the ecological services of Shanghai present a typical circle pattern, with the main urban area as the center and the importance level upgrades through the center downtown area to the outside area. Based on this feature and the current situation of Shanghai, the city area is roughly divided into four main functional areas as shown in Figure 10 (ecological bottom line area, ecological coordination area, ecological conservation area, and center construction area). The center construction area is the main construction area of the city, which is unimportant ecological land. This area is mostly used for commercial purposes (such as commerce centers, office buildings, and residential quarters) and lacks enough green space for local residents (lower than per capita public green area as reported in the government report of the eleventh five-year plan). The ecological coordination area is usually the boundary area surrounding the center construction area, which is mainly used as a coordination area for recreation and leisure for residents and ecological protection (such as emergency evacuation and flood diversion). The ecological conservation area is the basic ecological space of Shanghai and plays an important role in water and soil conservation, flood regulation and storage, wind and typhoon resistance, and habitat maintenance. Human activities, such as construction activities for commercial purpose, should be limited in an ecological conservation area to protect its ecological functions. The ecological bottom line area is an extremely important area of the ecosystem. This area has a significant role in an ecosystem. The area provides services such as water supply and biodiversity protection. Therefore, the ecological bottom line area should be protected.

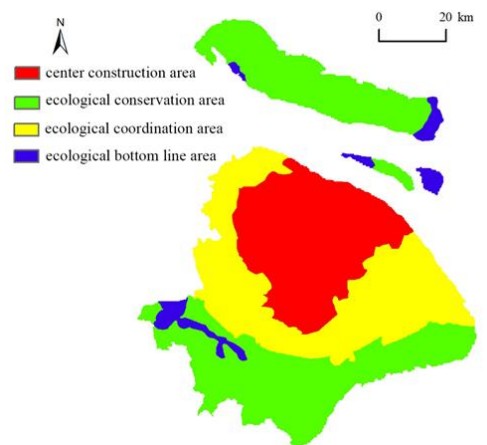

**Figure 10.** Ecological functional area in Shanghai.

### 3.2. Evaluation and Planning of Ecological Land Use at the Town (Meso) Scale

In the mesoscale study of ecological land use, towns with the areas of 50~200 km$^2$ are usually chosen. In this paper, Liantang Town was chosen in the mesoscale study. As shown in Figure 11a, the macroscale ecological land-use planning plays a guiding role in mesoscale, including the division of the main functional areas and the implementation of overall control indicators. In the main function division at the macroscale, it can be seen that most of the area of Liantang Town is defined as the ecological conservation area, while a small part of the town in the east is defined as the ecological bottom line area, which is located in the upstream water source protection area of Huangpu River. In Figure 11b, most of the area of Liantang Town belongs to important ecological land. A small proportion of the town area belongs to extremely important land, which is also mainly in the drinking water source protection area. The construction land of the town belongs to the unimportant ecological land.

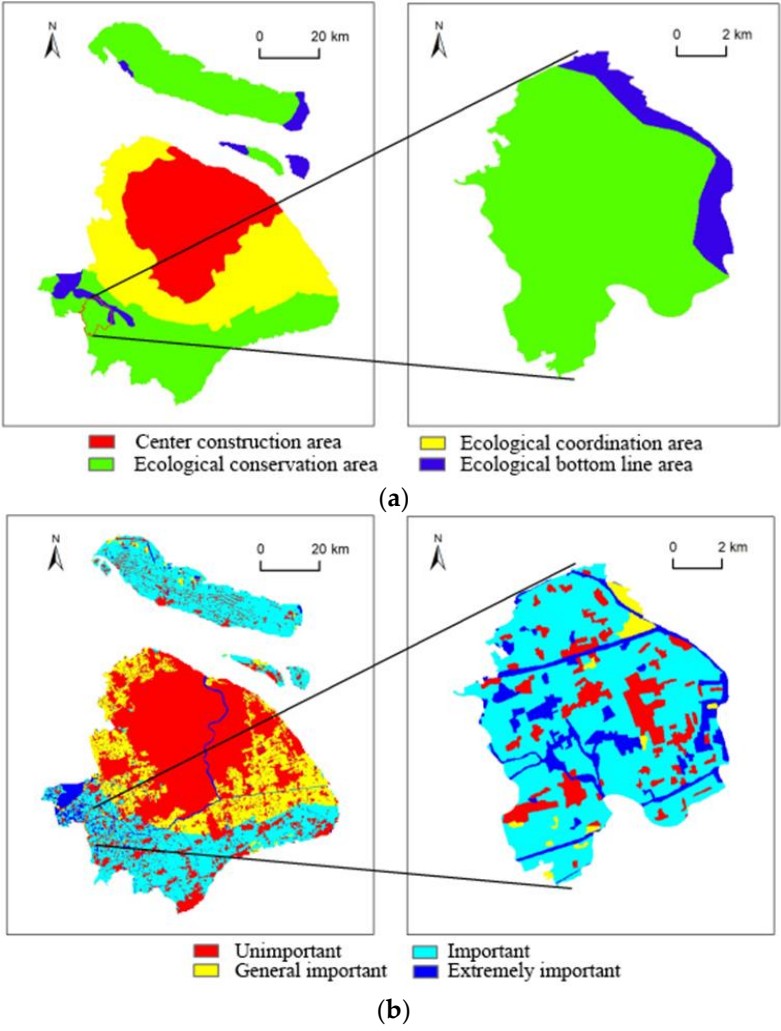

**Figure 11.** Macroscale guidance to the ecological functional zoning and importance levels of Liantang Town. (**a**) ecological functional zoning; (**b**) importance levels.

The current land use of Liantang Town is shown in Figure 12. The main land use of the town is arable land, construction land, and water area, which accounts for 68% of the total area. The arable area covers ~40% of the town. The construction land covers only 18% of the total area. The water area covers 21% of the area, which is relatively large. To evaluate the ecological services of Liantang Town, four indicators are selected, including water production capacity, habitat quality, carbon sequestration capacity, and grain production capacity. The results are shown in Figure 13.

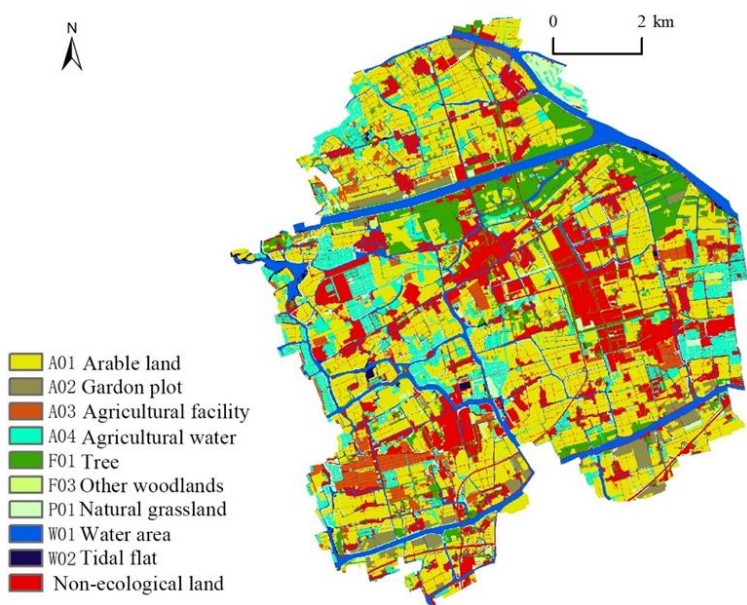

**Figure 12.** The land-use status of Liantang Town in 2018.

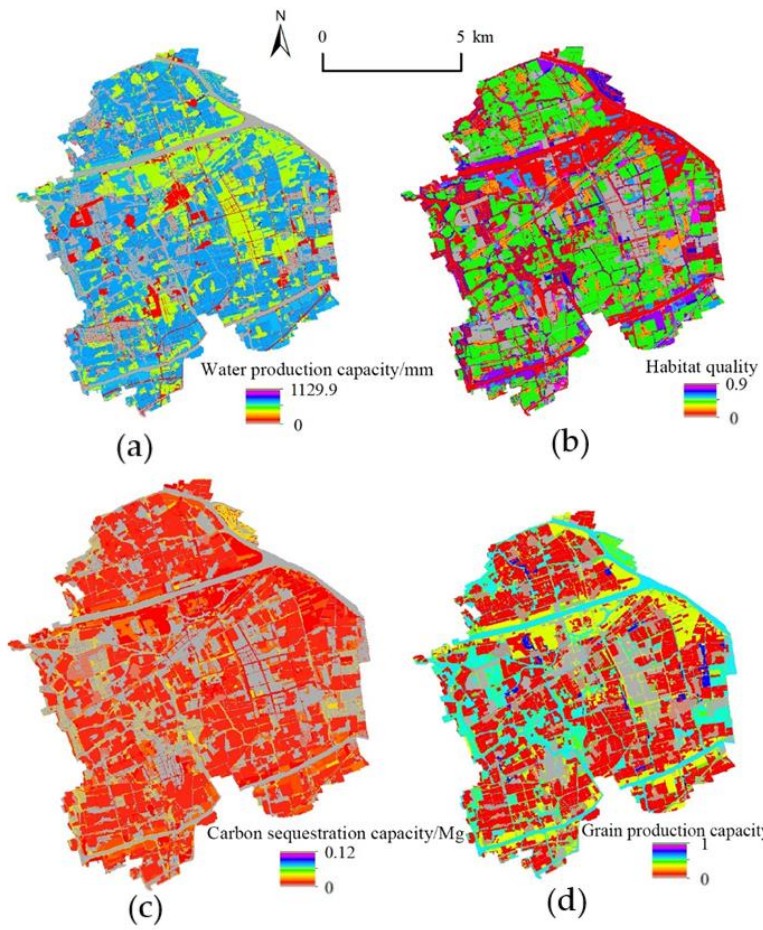

**Figure 13.** Spatial distribution of key ecosystem services of Liantang Town in 2018. (**a**) The water production capacity; (**b**) Habitat quality; (**c**) Carbon sequestration capacity; (**d**) Grain production capacity.

The water production service of Liantang Town (Figure 13a) displays obvious regional differences. The nonecological area has high water production capacity, because precipitation can easily turn into a

runoff on the impervious layer. Forest land has low water production capacity because of low water yield and rainfall interception. The water area also has low water production capacity because of the high evaporation. Also, the water area can effectively contain the water source, which can control floods and resist droughts.

The habitat quality of Liantang Town is shown in Figure 13b. The value of habitat quality indicates the habitat's ability to resist the influence of stresses. As seen in the figure, the nonecological land has relatively low habitat quality, as the area is highly influenced by human activities. Water area and forest land in Liantang Town have relatively higher habitat qualities because those areas are less influenced by human activities.

As shown in Figure 13c, the ecological land with the best carbon sequestration service is forest land, which provides more than 60% of carbon sequestration in less than 10% of the area of Liantang Town. Grassland also has good carbon sequestration service, while arable land has the lowest carbon sequestration ability. The land in Liantang Town is mainly arable land, which limits the carbon sequestration capacity.

The land with the highest grain yield in Liantang Town is arable land, followed by the water area. Notably, the water area provides 17% of the grain yield, which is related to the large water area and large amounts of fish ponds. According to the evaluation and analysis of grain production capacity, Liantang Town is a typical agricultural production town, along with the aquaculture industry.(Table 18)

**Table 18.** The ecosystem services of different land types in Liantang Town.

| Land Type | Land Structure | | Habitat Quality | Grain Production | | Carbon Sequestration | | Water Production | |
|---|---|---|---|---|---|---|---|---|---|
| | Area/km² | % | index | index | % | Mg | % | Depth/mm | % |
| Arable land | 37.44 | 39.64 | 0.41 | 0.98 | 68.1 | 0.01 | 24.5 | 882.49 | 47.5 |
| Garden | 2.85 | 3.01 | 0.75 | 0.50 | 2.5 | 0.02 | 2.49 | 912.35 | 3.7 |
| Forest land | 9.67 | 10.24 | 0.89 | 0.34 | 5.9 | 0.11 | 63.4 | 802.26 | 11.2 |
| Grassland | 5.73 | 6.07 | 0.70 | 0.43 | 4.5 | 0.03 | 9.65 | 873.50 | 7.2 |
| Water area | 19.69 | 20.85 | 0.90 | 0.51 | 17.2 | 0 | 0 | 19.17 | 0.5 |
| Residence | 11.59 | 12.27 | 0.11 | 0.04 | 0.9 | 0 | 0 | 1129.86 | 18.8 |
| Traffic land | 4.01 | 4.24 | 0 | 0.09 | 0.6 | 0 | 0 | 987.82 | 5.7 |
| Unused land | 3.47 | 3.67 | 0 | 0.03 | 0.2 | 0 | 0 | 1059.76 | 5.3 |

On the basis of the evaluation and analysis, four indicators were normalized and divided into four levels of importance (Figures 14 and 15), which are extremely important (0.85~1), important (0.75~0.85), generally important (0.65~0.75), and unimportant (0~0.65). The result shows that the ecosystem services of Liantang Town are mainly extremely important and important, which accounts for 57% of the total area, while generally important area accounts for 24% of the total area. The generally important and unimportant areas are located in the constructed area and its surroundings. After comparing the importance level of macroscale and mesoscale, the importance level areas mostly match. In macroscale, Liantang Town is mainly classified into extremely important and important ecological land as well. The further detailed importance division lists the generally important and unimportant ecological land for further management.

*3.3. Planning and Implementation of Ecological Land Use at a Community (Micro) Scale*

At the microscale, Changtian Community was selected as the research area. In this scale, field surveys and interviews with residents were carried out to obtain development wishes and planning requirements of residents. The layout of microscale land planning involves the interests of the public, which often determine the future direction of the social and economic development of the community. However, there are some uncertainties in the development goals of the communities. Therefore, multiple schemes/scenarios need to be compared to determine a relatively better practical scheme. Four schemes of different development goals and public demands are designed.

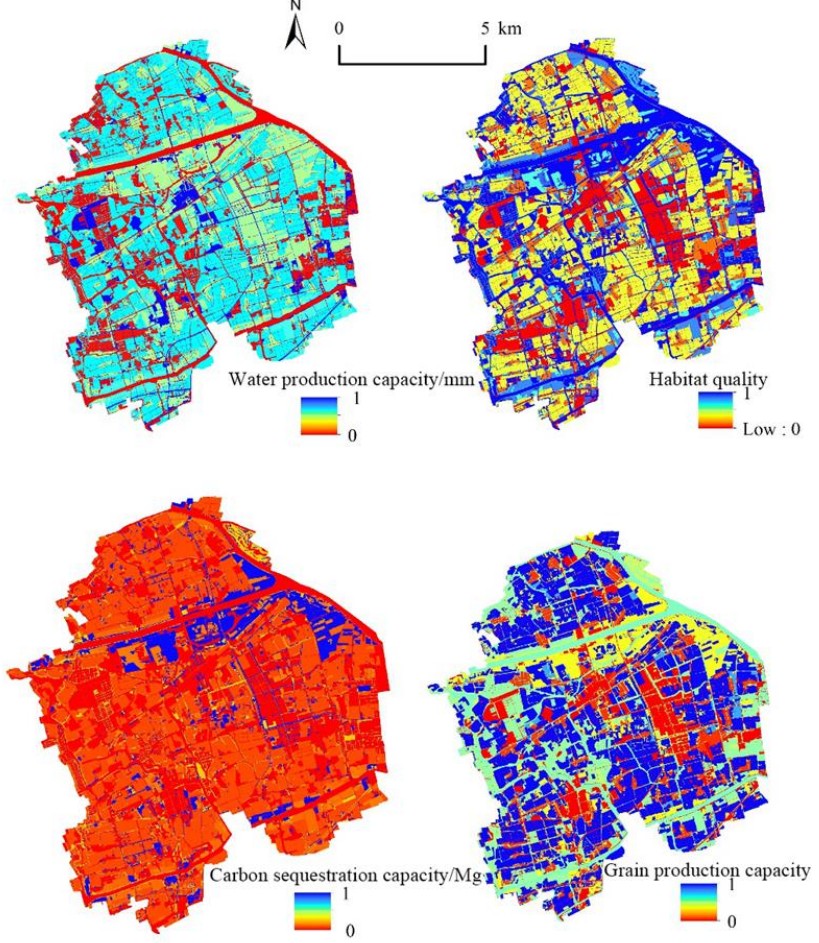

**Figure 14.** Normalization of indicators of ecosystem services in Liantang Town.

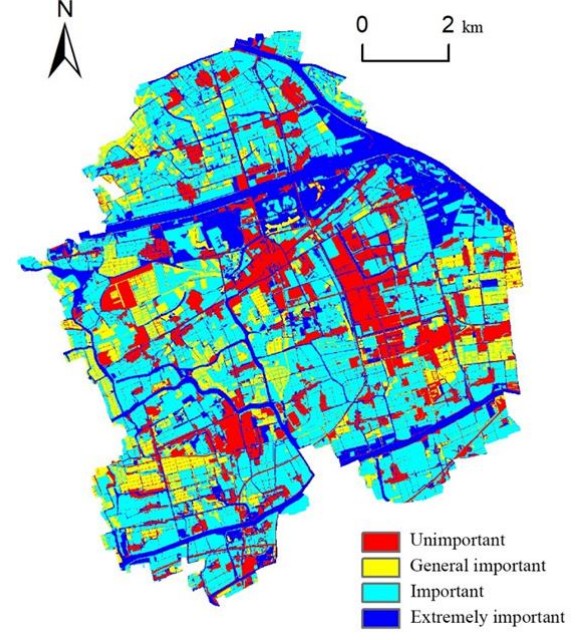

**Figure 15.** Importance level of ecosystem in Liantang Town.

The first scheme (Figure 16c) is an environmental improvement plan, which is aiming to renovate the ecological environment and build a clean community. According to the requirements of Shanghai environment improvement for rural areas, the environmental improvement plan is carried out with the purpose of beautifying the community environment and improving the living condition. According to this, this environmental improvement scheme contains five aspects of the key design: reduce the illegal construction land occupation and clear the main homestead area; regulate the waterfront ecological space and clear the riverside; widen the road in front of residence, reduce unnecessary hardened grounds and increase the green space; transfer the waste construction land into ecological land; keep the cultivated land, garden, forest land, and maintain their original layout.

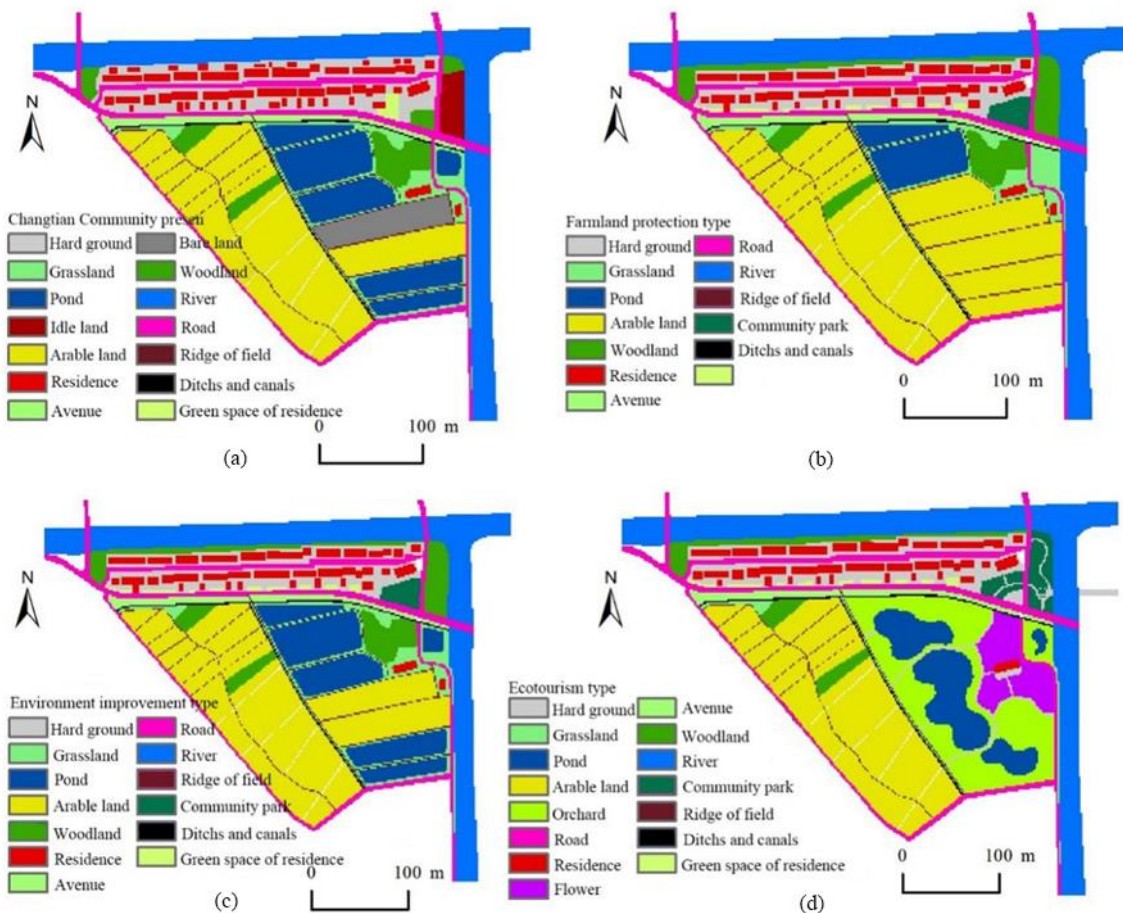

**Figure 16.** Land-use schemes of different objectives. (**a**) Changtian community; (**b**) environmental improvement plan, (**c**) farmland protection plan, (**d**) the ecotourism plan.

The second scheme (Figure 16b) is a farmland protection plan, which is mainly to improve agricultural function and protect cultivated land strictly. According to the requirements of macro and meso ecological land-use planning, Liantang Town bears great pressure on the farmland protection. Therefore, the main design principle is to protect basic farmland and increase the proportion of cultivated land area. The scheme contains four aspects of the key design: turn the current idle land into cultivated land to increase the ratio of cultivated land; turn the idle fish ponds into cultivated land or paddy field; carry out the high-standard farmland transformation to improve the farmland cultivation; carry out the basic comprehensive improvement of community environment to improve the living standard of residents.

The third scheme (Figure 16d) is the ecotourism plan, which aims to optimize the ecological environment and develop rural tourism. Liantang Town is famous for its red tourism and rural tourism. Changtian Community is close to the urban centralized construction area and has the location

advantage for the development of rural tourism. In the process of investigation, it is found that local residents also have the demand for developing rural tourism and the economy. The main design methods are as follows: renovate the natural rivers in the north and east, construct the waterfront leisure green space and form the waterfront ecotourism footpath; utilize the existing fish ponds, change the shape of them for tourism and sightseeing; adjust part of the cultivated land into orchards, increase the planting area of fruit forest, flowers and other crops, such as pear tree, peach tree, etc., and carry out picking events; optimize community public green space, improve the community's infrastructure supporting, and improve the residential environment to development homestay economy. Through the above designs, the ecosystem, tourism, and economics are integrated to achieve the development of a community economy on the basis of not damaging the natural environment.

To compare the present layout and three schemes, the land uses of different situations are calculated and listed in Table 19. The present land use of the Changtian Community is mainly cultivated land and water area, which account for 60% of the total area of the community. The environmental improvement plan aims to protect the environment and increase the area of arbor forest, avenue, grassland, and community park. The main purpose of the farmland protection plan is to improve the production capacity of agricultural products by increasing the area of arable land and improving grain productivity. The ecotourism plan is mainly to increase the main area of community parks, orchards and flowers, and emphasize the value of ecosystem services for sightseeing and entertainment.

**Table 19.** The areas of different land types in present layout and three schemes.

| Land Type | Present Layout (hm$^2$) | Farmland Protection Plan (hm$^2$) | Environmental Improvement Plan (hm$^2$) | Ecotourism Plan (hm$^2$) |
|---|---|---|---|---|
| Paddy field | 25.6 | 35.3 | 25.9 | 21.7 |
| Ditch | 1.0 | 0.9 | 0.9 | 1.0 |
| Orchard | 0 | 0 | 0 | 13.4 |
| Ridge | 1.9 | 1.7 | 1.7 | 0.9 |
| Flower | 0 | 0 | 0 | 4.1 |
| Pond | 12.0 | 8.4 | 10.6 | 8.1 |
| Arbor forest | 4.7 | 6.7 | 9.3 | 3.7 |
| Avenue | 2.0 | 1.8 | 3.9 | 2.0 |
| River | 19.5 | 17.3 | 19.9 | 19.5 |
| Community park | 0.0 | 0.9 | 1.6 | 2.2 |
| Attached green space | 0.4 | 0.6 | 2.2 | 0.7 |
| Grassland | 4.8 | 3.7 | 5.9 | 0.8 |
| Hard ground in residential area | 8.6 | 8.5 | 5.9 | 8.5 |
| Residential building | 6.0 | 6.1 | 4.8 | 5.2 |
| Road | 8.3 | 8.2 | 7.4 | 8.3 |
| Idle space | 1.5 | 0 | 0 | 0 |
| Bare land | 3.5 | 0 | 0 | 0 |

To solicit opinions from local residents and managers, questionnaires were developed and distributed. The questionnaire results show that residents support ecotourism plan > environmental improvement plan > farmland protection plan. The residents suggested two aspects for future development: improve the living environment and basic service facilities of the community; improve the tourism industry in the future.

The results from managers are different and they support the environmental improvement plan > farmland protection plan > ecotourism plan. There are five suggestions proposed. Firstly, arrangements of administrative tasks and basic land management policies of the state should be obeyed; difficult work of land function adjustment and coordination should be avoided in planning and design; basic farmland should be protected and pollution sources should be controlled to maintain and improve river water quality; community environmental improvement is a compulsory task, which needs to be actively implemented and completed; development of tourism industry in the community should be promoted.

Based on the opinions of residents and managers in Changtian Community, consensuses on future development can be formed:

1. Improve the community environment and dismantle illegal buildings;
2. Keep the current function of landscape and avoid changes of land function;
3. Increase the compound function of existing land and enhance the tourism value and economic value of the community;
4. On the basis of farmland and ecosystem protection, adjust some of the lands, change the crop types, and develop a characteristic economy.

To compare three schemes qualitatively, ecosystem service values are calculated and compared (Figure 17). As shown in Figure 15, supply service, regulating service, supporting service, and cultural service of three schemes and present land use are compared. Four service values of different schemes differ from each other. Therefore, considering one service value is far from enough. All aspects should be considered, so that its comprehensive value can be maximized and the community can serve better.

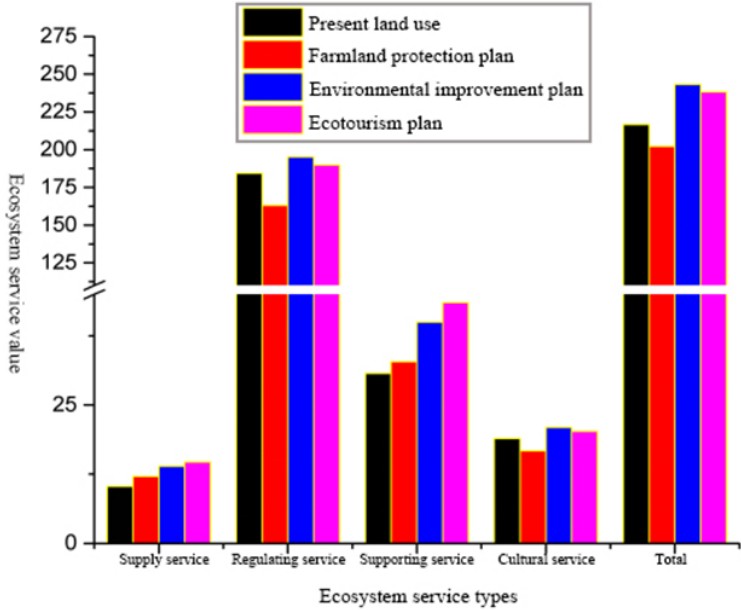

**Figure 17.** Comparisons of ecosystem service values under four scenarios.

Based on the comments from residents and managers of the community, an optimized scheme is proposed and the layout is shown in Figure 15. The optimized scheme is shown in Figure 17. The optimized scheme includes five aspects of the suggestions:

1. The optimized scheme takes environmental renovation and ecotourism into account to realize the balance between protection and economic development;
2. Delimit clear main functional areas to form functional blocks with different characteristics;
3. Keep the existing cultivated land area and improve the quality of cultivated land; keep the boundary and reduce the difficulty of land-use adjustment;
4. Utilize the land types such as garden and fish pond, increase the leisure tourism functions such as sightseeing, picking, recreation, etc., which have less impact on the environment;
5. Focus on the improvement of waterfront space, combine the ecological restoration with sightseeing and recreation functions, and enhance the cultural value of the waterfront space.

The optimized scheme (Figure 18) combines the present land use and future development, which includes environmental protection, farmland protection, and ecotourism. It suits the administration policy and the anticipation of residents.

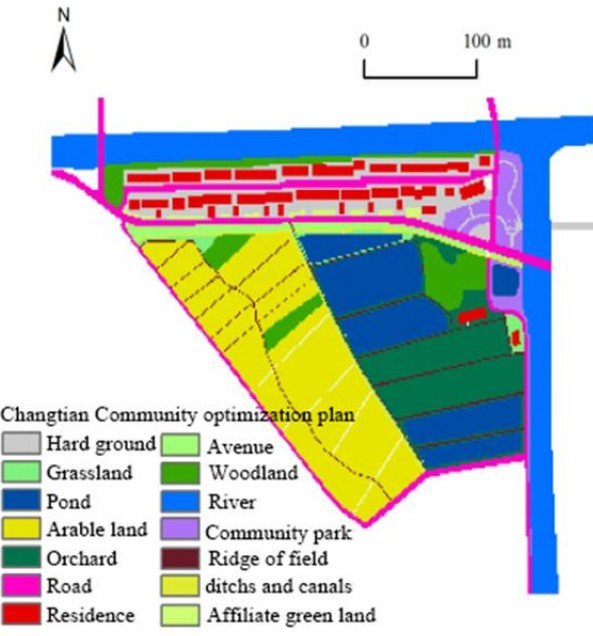

**Figure 18.** Optimized scheme of Liantang Town.

## 4. Discussion and Conclusions

The ecological land and its protection have been a hot spot of our society for years. In this paper, urban ecological land refers to the land/area that can directly or indirectly provide ecological service (such as environmental material supply, climate regulation, ecological support) to the city to protect and stabilize the regional ecosystem [25]. Currently, the concept of ecological land is not clearly defined in land management in countries other than China. However, the concept of ecology has been in land-type division and land management for years. In 1981, Rowe et al. used the techniques of aerial photographing to identify and classify land types based on the relevant theories of ecology [26]. In 1983, an American ecologist named Bailey took land as a composite ecological system and, in the process of classifying it, he integrated the land of ecological significance into a larger geographical unit and built connections with surrounding geographical units [27]. In 1990, Zonneveld from Holland discussed the spatial levels of ecological environment, land piece and land system based on the theory of landscape ecology [28]. Klijn and Haes constructed the classification system and standard of ecological land and elaborated the application value and significance of the classification system [29].

At present, the common land classification systems include LUCC (Land-use and Land-cover Change), Anderson of USGS (U.S. Geological Survey) and CORINE of the European Union. These systems are all based on the theories of ecology and geography and classify land types by geomorphic characteristics, ecological functions, land covers, et al. These studies have not concluded the unified concept of ecological land, but researchers focused on the land layout managements and ecological land divisions to evaluate the ecological suitability of land and explore the dynamic evolution process of vegetation cover of ecological land.

This study takes Shanghai as a case study and discusses the ecological land evaluation and management in three different scales. On the macroscale, a 30 m resolution remote sensing image of Shanghai and the InVEST model is used to interpret the main land-use situation and evaluate the importance level. In mesoscale, the 1 m resolution satellite remote sensing image of Liantang Town was used and four indicators (carbon sequestration capacity, habitat quality, water production capacity, and grain production capacity) were chosen to evaluate the ecological land of the town. In microscale, Changtian Community in Liantang Town was chosen as a case study. Multiple schemes were proposed and optimized after consulting with managers and residents.

The adaptive evaluation and planning of ecological land in Shanghai are the key process and result of this study. Previous land-use managements use one single method to classify and manage different lands, which would lead to the problems of adaptabilities in different scales and different regions. China is a country with extensive lands and one method is too simple to manage this country. Therefore, this strategy of adaptively managing ecological lands can meet the demands of ecology and economic development. The management of ecological land cannot be done in one process and it needed to be evaluated and decided by different decision-makers. This is, far and away, the most important point of ecological land management.

**Author Contributions:** The following statements should be used "conceptualization, W.J. and Y.C.; methodology, W.J. and Y.C.; software, W.J.; validation, W.J. and Y.C.; formal analysis, W.J.; investigation, W.J.; resources, W.J. and Y.C.; data curation, W.J.; writing—original draft preparation, W.J.; writing—review and editing, J.X.; visualization, W.J. and J.X.; supervision, Y.C. and Z.L.; project administration, Z.L.; funding acquisition, Z.L. All authors have read and agreed to the published version of the manuscript.

**Funding:** This work was supported by grants from the National Science Foundation of China (41773004), China National Key R&D Programmes (2016YFC0502701-2), the Jiangsu Provincial Key Laboratory of Radiation Medicine and Protection, the Priority Academic Program Development of Jiangsu Higher Education Institutions (PAPD).

**Conflicts of Interest:** The authors declare no conflict of interest.

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
