# Peer review of "Ecological Land Adaptive Planning in Macroscale, Mesoscale, and Microscale of Shanghai"

_sustainability, doi:10.3390/su12052142_

Round 1

Reviewer 1 Report

The reviewed paper pays attention the process of urbanization connected with the cities’ development and changes with ecological land. However, the discussed issue is extremely important in the contemporary world, the content of the paper and employed methods let us to ask many questions.

There is no aim of the paper. It is difficult to consider or analyze any problem without clear aim. Study area. Authors proposed three steps research approach: the first – macro-level, the second – the meso-level and the third – the micro level. The first level included Shangai city. I do not have any comments here because its selection is explained (the largest city in China). The meso- and micro-levels are not argued enough. We do not know why and how research areas at every level were made. What is more, in the chapter 2.1., there is not precise information that selected area represents meso-level. Results and discussion. I am not sure if the word “discussion” is proper in this chapter. There is no discussion. The results, however interesting, do not provide any explanation why some changes have been observed in Shanghai. Such a pity. In the presented form it rather looks like a report but not a research paper. In the section 3.3. the authors mention some questionnaire survey but they do not explain who were the respondents, what about the sample, in what conditions the survey was conducted. There is no discussion. We as readers do not have any information if observed changes and proposed solutions are really unique or rather typical for Chinese cities. The literature review is limited to 14 references. In fact, there is no typical literature review with the description of research problem in the paper. In results, the conclusions are very poor.

Author Response

We have rewrite the manuscript based on the suggestions.

In chapter 2.1, a new map was added. And at this part, data and method was rewrite to further explain the aim of this paper. Different level of study area were given in more details.

Result and discussion part was rewrite for better explaining the aim of the paper. For the part of 3.3, the discussion was added at part 4.

Reviewer 2 Report

While facing the rapid urbanization progress widely seen in China, the management and planning of ecological land meets many problems. This paper proposes a three-layer framework (macroscale, mesoscale and microscale) to evaluate, classify and plan the ecological land in Shanghai. Generally, it sheds some lights on the way of coordination among different administrative levels, the residents and managements who represent stakeholders holding different priorities. However, two obvious shortcomings hamper the authors from completely delivering their thoughts.  

The structure arrangement is problematic, especially the section 1, 2 and 4. Section 1 broadly introduces the background of the research, but fails to sharpen the focus of this study, i.e. what is the status quo of the research on urban ecological land management, what are the innovative ideas in this paper different from existing research and so on. Section 2 is too general. The readers can hardly get any clues for the selection process of the four indices in the macro-and meso-level, the calculation and normalization of the indices, source of the three schemes, and the design of the questionnaires at the micro-level. Section 4 is largely vague. It fails to base the discussion upon the results from the previous section. The paper proposes a three-layer analytic framework of the ecological land management and provides detailed analysis in the section 3. This is the highlight of the research. However, the linkage and the integration of the three layers are much less touched on. Therefore, this framework so far looks quite disaggregate and cannot sufficiently support the authors’ idea of adaptive planning.

Besides the above issues, there are also trivial problems, such as:

How to read the table 2, i.e. what is the transfer direction? In L. 145, the concept of “compromise of arable land” is not accurate. Do the authors refer to the quantity or the quality? Table 3 is not referred in the manuscript. Table 4 lacks the footnote for the meaning of the double star symbol. The correlation between land use types and habitat quality is missing. I suggest to replace the column “average” by “change ratio” in table 5 to reflect how the changes of habitat qualities react to the trend of urbanization. Table 6 is not referred in the paper. I suggest to add analysis on reasons of the worsening habitat quality of each kind of land use type along with the years. Brief introduction of the significance of the selected indicators, for example the depths and yields in L.173 can help readers understand the authors’ intentions better. In L.195, “the water capacity of the soil” is not accurate. In L.196, “the water supply of nature” is not correct. How to keep the consistency between the importance level and ecological functional zone, as the boundaries of Fig. 7 and 8 are not identical? I suggest to insert figures correlating the importance levels with land use types, so as to better reflect the implications of urbanization on ecosystem services. Therefore, the analysis ranging from L.237 to L.244 is arguable. To my understanding, the classification of the importance level and the following ecological functional zone is the result of land use change caused by urbanization. Therefore, when discussing the protection of those important areas, we need to touch on the root reason instead of simply advocating limit the construction. Sentence “as seen… low interference” in L.275-277, the phrase “The land Liantang Town” in L. 281, sentence “…to determine a relatively better for implementation” have grammatical errors. What does “red tourism” in L. 334 refer to? Fig.14 is missing. The title of Tab.13 is missing. In L. 387, it should be Fig. 16. Sentence “…food supply is chosen…area in Shanghai” in L. 412-413 is grammatically problematic.

Author Response

Section 1, 2 and 4 has been revised as suggested and the changes can be seen in the new revised edition. Generally, the main focus of the research is introduced in section 1, along with the difficulties and innovations. The detailed methods are introduced in section 2. The model mainly used in this study are introduced in this section and the indicators and their corresponding data used in different levels are also introduced. At the meantime, the detailed technical process is introduced in the last part of section 2. Section 4 is also revised to provide the discussion on this study.

Besides the above issues, there are also trivial problems, such as:

How to read the table 2, i.e. what is the transfer direction?

As shown in the table below, c is the percentage of B turns into A during the study period.

Land type

A

B

A

a

b

B

c

d

In L. 145, the concept of “compromise of arable land” is not accurate. Do the authors refer to the quantity or the quality?

The compromise of arable land means the loss of the arable land.

Table 3 is not referred in the manuscript.

Table 3 (now Table 9) is actually referred in the previous manuscript but I forgot to add the reference. The reference is added in the revised manuscript.

Table 4 lacks the footnote for the meaning of the double star symbol.

The footnotes for the star and double star symbol have been added.

The correlation between land use types and habitat quality is missing.

There is no correlation between land use types and habitat quality in the first place. However, there is habitat quality index of different land types in Shanghai (Table 12).

I suggest to replace the column “average” by “change ratio” in table 5 to reflect how the changes of habitat qualities react to the trend of urbanization.

Revised as suggested.

Table 6 is not referred in the paper. I suggest to add analysis on reasons of the worsening habitat quality of each kind of land use type along with the years.

Shanghai has been experiencing high-speed changes in last decades. Due to large number of human expansion and economic development, the construction area expands and takes over other ecological lands, such as forest area and grassland. The detailed reasons of the worsening habitat quality of each kind of land use is too complicate to know.

Brief introduction of the significance of the selected indicators, for example the depths and yields in L.173 can help readers understand the authors’ intentions better.

The introductions of indicators are added in the methods (section 2). (I’m afraid that line numbers do not match in our versions and the problem occurs in the following questions. But I tried my best to revise the paper and hope it meets your demands)

In L.195, “the water capacity of the soil” is not accurate. In L.196, “the water supply of nature” is not correct.

“The water capacity of soil” has been changed into “the water capacity of the underlying surface” and “the water supply of nature” has been changed into “the water cycle in nature”.

How to keep the consistency between the importance level and ecological functional zone, as the boundaries of Fig. 7 and 8 are not identical? I suggest to insert figures correlating the importance levels with land use types, so as to better reflect the implications of urbanization on ecosystem services. Therefore, the analysis ranging from L.237 to L.244 is arguable. To my understanding, the classification of the importance level and the following ecological functional zone is the result of land use change caused by urbanization. Therefore, when discussing the protection of those important areas, we need to touch on the root reason instead of simply advocating limit the construction.

The importance levels are the different ranges of the normalized results of four indicators and the calculation is introduced in section2. The division of ecological functional zone is based on importance levels, the main function of the present area and local management rules/laws. Therefore, the boundaries of two figures roughly matches. The rough evaluation provides the importance and the main ecological function in macro scale, which further provide evidence in meso and micro scale management.

Sentence “as seen… low interference” in L.275-277, the phrase “The land Liantang Town” in L. 281, sentence “…to determine a relatively better for implementation” have grammatical errors.

The sentence has been revised as “as seen in the figure…because those areas are less influenced by human activities.” The phrase has been revised as “the land in Liantang Town”. The sentence has been revised as “…to determine a relatively better practical scheme”

What does “red tourism” in L. 334 refer to?

Red tourism is a specific Chinese traditional tourism type, which is mainly based on the Chinese revolution in modern times.

Fig.14 is missing.

A number mistake. Sorry for that.

The title of Tab.13 is missing.

The title is added.

In L. 387, it should be Fig. 16.

Sorry for the mistake. A few figures has been added in and the figure numbers are different now.

Sentence “…food supply is chosen…area in Shanghai” in L. 412-413 is grammatically problematic.

The sentence has been revised as “the grain production supply were chosen as one indicator because Liantang Town is a typical agricultural production area in Shanghai.”

Reviewer 3 Report

This study was aimed at looking ecological assessment using InVEST ecosystem management tools in Shanghai. Although this study brought several interesting findings on about the ecological crisis-that are needed to be addressed in the city, I fell the manuscript needs to be revised thoroughly. I have listed my observation below.

TITLE: this should be changed. This does not tell readers anything. I recommend authors find a meaningful title that fits your research objectives and findings. The abstract is not clear enough, I recommend authors rewrite it; shortly describe your research objectives, employed methods and wrap up with findings In introduction, citation numbers are not properly arranged. Introduction is not well developed, more focused should be given issues related to ecological crisis in the study area and what studies are previously conducted In study area, add a map showing the study area in China. Also expand it bit more adding population, economic development, and current urban growth trends The data and methodology are incomplete. InVEST model was used, but I don’t see any details about different sub-models, data preparation and model calibration process. It seems four models were executed such as habitat quality, carbon, water and food supply without provided further information about how these models work for what purposes. I wonder how the authors calculated carbon pools such as; above ground, below ground, soil and dead matter carbon, I guess these are the required input along with LULC. Please clearly states all models, data used for each model and their details estimation and execution process in Methodology section. Move all discussion about models from Results section to Methodology section. In results section, present only findings. In table, findings are presented with district or other administrative level. But this doesn’t tell us where they are located. I suggest providing all maps with district/ stated administrative level boundary with proper labels. In map legends are not clear. What is the unit? Please label each legend of map properly.

Author Response

 Thank you for the honest reviews and we have revised the paper thoroughly according to your advice. The title has been changed and so do the rest part of the manuscript. The study area, methods and data part (section 2) has been mainly revised to state the methods and data used in this research. The indicators used in different scales and their calculation/ methods are added. Some detailed information is also added in this part, such as the map of Shanghai and the administration division of Shanghai. The revisions can be seen in the new manuscript.

Round 2

Reviewer 1 Report

I accept the paper on the presented form .

Author Response

The aim of this paper is to provide a systematic method to manage ecological land in China. We chose Shanghai as the study area because the city is a typical example of overpopulation and urbanization and the ecological land area is reducing over the years. The scales are chosen as the administration level in China for practical reasons. In the result section, the changes in Shanghai has been added as required. The questionnaires are answered by residents and managers of the community as explained in Section 2.2.3. The questionnaire mainly asks their opinions about different management plans we proposed and we did not present the original questionnaire because it is in Chinese. The references in this paper are limited because the results cannot be compared with other results. It is a new management method we proposed based on the current management.

Reviewer 2 Report

I am much appreciated of the authors’ great efforts on the manuscript improvement. Now the structure of the paper looks appropriate and some technical errors have been corrected. However, in the current shape, there are still several questions left to be answered, especially an obvious flaw lying in the framework, i.e. the second big issue proposed in my previous comments. Please consider how to properly illustrate the coordination of the indicator selection for the three levels, and how the thought of adaptive planning is integrated. In my opinion, the discussion section is more kind of the explanation on the analytical framework, which should be put in section 2.2.4.

Other questions:

In the previous comments, I suggest the analysis ranging from L.237 to L.244 (in the orginal version) is arguable. How to solve this? Authors have replied “The division of ecological functional zone is based on importance levels, the main function of the present area and local management rules/laws. Therefore, the boundaries of two figures roughly matches”, which doesn’t have corresponding information in the manuscript. 68-72 (in the new version) needs citation to support authors’ judgement. More explanations and citations are needed to justify L.109-110, L.149-151, L.164-165(in the new version) 514-515(in the new version) is grammatically problematic.

Author Response

The framework of this paper has been adjusted as suggested, which can be seen in the new version of the manuscript and we put the discussion section into the analytical framework.

Sorry that L237-244 in my manuscript are totally different and we could not locate the corresponding descriptions. We added description in the new manuscript (Based on this feature and the current situation of Shanghai, the city area is roughly divided into four main functional areas (ecological bottom line area, ecological coordination area, ecological conservation area and center construction area)).

L68-72 (Hope we are looking at the same version) are supported by citation No. 1~5.

We ran through the whole manuscript to rule out the grammar problems (because the lines doesn’t match, sorry about that, again).

Reviewer 3 Report

Authors did not improve the manuscript accordingly. It is lack of novelty, obscurity in employed methodology and overall poor writing and formatting  .

The abstract is poorly written, hard to grasp the idea what the authors wanted to express. For example; the original ecosystems, a single discipline, the carrier and lots more. These are not academic and research sound to be written in abstract. Abstract is also vague, filled in description without given any outcomes. texts are not cited properly though I instructed in first review. For example; L38-41, L79-92 in L112: several parameters from previous research, what are they? Authors did not provide how Table 1, Table 2, comes from. If this has been done previously, I wonder what the novelty of this study is? I wonder what the previous studies were. I expect a broad literature review on previous studies and how this study contributes differently from others I recommend every map should come with district level administrative boundaries with labeling, but, unfortunately the authors did not follow the instructions L401: the authors did not mention about field survey in methodology section, and how this was conducted and relate with this research. Overall, this manuscript doesn’t lead the readers in the next step.

Author Response

The abstract has been revised in the new manuscript. We tried to be more specific this time.

The parameters provided in the paper are the parameters used in the model. We provide these parameters in this paper, hoping they might help other researchers someday.

The references of table 1 and 2 are added in this version.

The method of ecological land management proposed in this paper is new, which can be promised. However, the researches/papers we studied are mostly in Chinese. We try not to cite all these papers and give a brief introduction in this paper.

We have revised this manuscript thoroughly under the suggestions of all viewers and hope you will like this version.

Round 3

Reviewer 2 Report

The framework of this paper has been adjusted as suggested, which can be seen in the new version of the manuscript and we put the discussion section into the analytical framework.

It is a good direction to focus on the illustration of the proposed multi-scale systematic adaptive ecological land planning method, i.e. section 2.2.4. However, the new structure for me is more kind of a list of arguments to answer my question “how to properly illustrate the coordination of the indicator selection for the three levels, and how the thought of adaptive planning is integrated”. Therefore, the narratives look organized in a rush and the information is fragmented. I suggest the authors to reconsider the role of this section in the whole paper and revise in a systematic way.

In addition, since authors move the discussion section in to section 2.2.4, there is no discussion at all in the new version. Actually, the reflection on the applying of this proposed method should be involved in the discussion part and furthermore, there can be comparison between this new method with other alternative methods.

Sorry that L237-244 in my manuscript are totally different and we could not locate the corresponding descriptions. We added description in the new manuscript (Based on this feature and the current situation of Shanghai, the city area is roughly divided into four main functional areas (ecological bottom line area, ecological coordination area, ecological conservation area and center construction area)).

Here I attach my comments and the original manuscript here for your reference.

My comments: Therefore, the analysis ranging from L.237 to L.244 is arguable. To my understanding, the classification of the importance level and the following ecological functional zone is the result of land use change caused by urbanization. Therefore, when discussing the protection of those important areas, we need to touch on the root reason instead of simply advocating limit the construction.

Original manuscript:

In this area, more green spaces should be created for recreation and leisure. The ecological coordination area is usually 237 the boundary area around the center construction area, which is mainly used as coordination between 238 urban development and ecological protection. The ecological conservation area is the basic ecological 239 space of Shanghai and plays an important role in water and soil conservation, flood regulation and 240 storage, wind and typhoon resistance and habitat maintenance. Therefore, constructions should be 241 limited and the ecosystem should be protected in the ecological conservation area. The ecological 242 bottom line area is an extremely important area of the ecosystem. The ecosystem in this area should 243 be protected, including water supply, biodiversity protection and other key ecological elements.

L68-72 (Hope we are looking at the same version) are supported by citation No. 1~5.

Since L68-72 are basically a conclusive judgement on existing research, a reviewing process which leads to this conclusion is missing. As authors reply that their arguments are supported by citation NO.1-5, I suggest to insert the literature review part based on citation 1-5.

We ran through the whole manuscript to rule out the grammar problems (because the lines doesn’t match, sorry about that, again).

There are still minor mistakes. For example, in L. 226, “…is a effective technique…” should be “…is an effective technique…”

Author Response

Thank you for your suggestions. We are trying to show readers how we process our information and data and how we proposed the final plan of the community, which is the highlight of the whole paper. As you can see in section 2.2.4, the study uses the downscaling method (from macro scale/city scale to micro scale/community scale). The macro scale is mainly used to provide an overall evaluation and planning in Shanghai city. The city scale management is decided by the government and supported by laws and regulations. Therefore, in the smaller scales’ planning, we should follow the overall planning and official framework. In the meso scale, we try to subdivide the ecological service land to provide constraint information of ecological control condition for the next scale management. In the micro scale, we proposed several future land management plans by following the ecological land evaluation from macro and meso scales. We combined the assessment and feedback from local residents and managers of the community and proposed an adaptive plan for the community. The final plan can be used in the future because it follows the overall plan of Shanghai city, provides basic ecological services and is also profitable from farming, fruit and tourist industries combining with rural tourism. In this paper, we fail to provide details of laws and regulations of Shanghai (we did a lot of researches on that) and also skip some details because of the limitation of the length and we truly feel sorry for that. Because in China, different cities have different management plans that are supported by local laws and regulations. We want to introduce our method of ecological land plan and hope our method will inspire some researchers in the future.

We add a new discussion section in the new manuscript. And about the grammar mistakes, we located the mistakes and revised them. Again, thank you for your patience and good suggestions.